# Capture of associated targets on chromatin links long-distance chromatin looping to transcriptional coordination

Ryan J. Bourgo[1], Hari Singhal[1] & Geoffrey L. Greene[1]

Here we describe a sensitive and novel method of identifying endogenous DNA–DNA interactions. Capture of Associated Targets on CHromatin (CATCH) uses efficient capture and enrichment of specific genomic loci of interest through hybridization and subsequent purification via complementary biotinylated oligonucleotide. The CATCH assay requires no enzymatic digestion or ligation, requires little starting material, provides high-quality data, has excellent reproducibility and is completed in less than 24 h. Efficacy is demonstrated through capture of three disparate loci, which demonstrate unique subsets of long-distance chromatin interactions enriched for both enhancer marks and oestrogen receptor-binding sites. In each experiment, CATCH-seq peaks representing long-distance chromatin interactions were centred near the TSS of genes, and, critically, the genes identified as physically interacting are shown to be transcriptionally coexpressed. These interactions could potentially create transcriptional hubs for the regulation of gene expression programmes.

[1] Ben May Department of Cancer Research, University of Chicago, Chicago, Illinois 60637, USA. Correspondence and requests for materials should be addressed to G.L.G. (email: ggreene@uchicago.edu).

Chromatin architecture is a key regulator of many aspects of cell biology, including gene transcription, DNA repair processes, DNA replication and long-term processes such as X-chromosome inactivation[1]. A host of transcription factors, enzymes, scaffolding proteins and other factors ensure that the local chromatin architecture is a dynamic environment that directly and indirectly regulates the complex cellular processes noted above[2]. This complex network of chromatin architecture is largely composed of the non-protein-coding genome. Estimates of the true functional percentage of our genome range from 10% (ref. 3) to as much as 80% (ref. 4), but despite this discrepancy, we are certain that only a small fraction of the genome has been evolutionarily conserved, largely in those protein-coding regions. Perhaps not surprisingly, transcriptional enhancers are disproportionately common among evolutionarily conserved non-protein-coding sequences[5].

Transcriptional enhancer regions are hubs of transcription factor binding, and are thought to underlie a significant portion of the tissue-specific expression of many gene targets[5–7]. Whereas gene promoters are enriched for H3K4me3, these enhancer regions contain almost exclusively the mono-methylated version of H3K4 (H3K4me1). In addition, it has been found that most active enhancers are characterized by the increased presence of H3K27ac. The epigenetic enhancer signature[8] has greatly contributed to the genome-wide prediction of transcriptional enhancer sites. It has become increasingly clear that the majority of transcriptional enhancers are not located within or directly adjacent to the genes they modulate, but are typically located at great linear distance. Despite the long-range linear distance (in base pairs) between two interacting loci—in many cases, hundreds or thousands of kilobases—the prevailing model is that of a physical confrontation between the two sites. This requires long-distance genomic looping to occur, whereby two linearly distant genomic loci come into close proximity, and are held together by complexes of proteins and transcription factors[9]. While the looping mechanism of DNA–DNA interaction has been postulated for nearly four decades[10], it has been notoriously difficult to study.

The current benchmark assay for detecting long-distance chromatin interactions is Chromatin Conformation Capture (3C)[11], as well as a number of derived techniques discussed in more detail in the discussion. Critically, these assays rely on random end ligation at very low DNA concentrations after restriction digestion, which reduces assay reproducibility and results in significant data loss[12]. Ultimately, 3C-based assays are easy to corrupt, difficult to troubleshoot and are impractical to the average research laboratory[13]. Thus, the field of chromatin interaction is lacking in tools to facilitate mechanistic understanding of this process.

To overcome the current technological limitations that constrain the field of DNA–DNA interaction research, we have developed a streamlined, novel protocol named Capture of Associated Targets on CHromatin (CATCH). Similar to traditional 3C, CATCH relies on chemical crosslinking to capture naturally occurring nucleic acid–protein interactions. However, CATCH relies on an unbiased sonication approach to shear DNA, similar to ChIP assays. A genomic locus of interest is enriched through hybridization and subsequent purification using a complementary biotinylated oligonucleotide. Owing to the use of formaldehyde crosslinking, this purifies both the targeted DNA sequence and any interacting nucleic-acid segments. Finally, following de-crosslinking, the resulting DNA sample is subjected to PCR or sequencing to identify interacting fragments. The CATCH assay has significant and extensive list of advantages over 3C-derived technologies, which is considered at length in the discussion section below.

In the present study, we test the effectiveness of CATCH by interrogating a downstream enhancer of the human SIAH2 gene, which had been previously analysed using chromatin interaction analysis by paired-end tag sequencing (ChIA-PET). SIAH2 (3q25.1) is an E3 ubiquitin ligase whose upregulation correlates with ER activation and has been linked to poor outcome in breast cancer patients[14,15]. Currently, SIAH2 gene transcriptional control is poorly understood: the only confirmed genomic loop within SIAH2 occurs between an intronic oestrogen response element (ERE) and downstream ERE[16]; however, multiple ER-binding sites are present within and around the gene. In order to resolve the interactions involved in SIAH2 regulation, as well as attempt to recapitulate the previously demonstrated looping events around SIAH2, we perform next-generation sequencing after CATCH of the SIAH2 downstream enhancer. In addition, we demonstrate the reproducibility of CATCH using distinct pulldowns near the SIAH2, EIF4A1 and MYC genes. These experiments also show that CATCH-seq peaks are overwhelmingly found overlapping with enhancers (H3K4me1 and H3K27ac enriched) and oestrogen receptor (ER)-binding sites. Finally, these experiments reveal unique subsets of physically interacting gene promoters that are shown to be transcriptionally coexpressed over thousands of data sets using the SEEK search system[17].

## Results

**CATCH identifies long-distance genomic interaction.** A flow-chart representing the CATCH methodology and process is shown in Fig. 1. The use of sonication is a relatively unbiased method of DNA shearing that has obvious advantages. As a proof-of-concept for CATCH sonication and fixation efficiency, we chose to target an intronic region of the SIAH2 gene that had previously been shown to interact with a region downstream of the gene[16]; this intronic region is an established ER-binding site and was designated ERE[B]. The human SIAH2 gene consists of two exons and a single intron located on the minus strand of chromosome 3; ERE[B] is located within the intron, ~6.8 kilobases from the SIAH2 promoter. Because unsheared DNA or over-fixed samples could provide false-positive results, random genomic loci were also tested for interactions within SIAH2. The tested target or random loci ranged in distance between 1.1 and 19 kilobases from ERE[B] (Supplementary Fig. 1a). Probing for the presence of each locus in the pool of gDNA that was subjected to the hybridization and capture steps resulted in the amplification of each locus, as expected; this result demonstrated that formaldehyde fixation and sonication do not destroy or bias the availability of genomic loci (Supplementary Fig. 1b). Importantly, without formaldehyde fixation, it was possible to pull down additional loci with ERE[B] only if the sample was incompletely sonicated (Supplementary Fig. 1b), because of the proximity of the two loci on the linear genome. This result demonstrated that any interacting loci seen with the addition of formaldehyde could be interpreted as transient physical interactions that require static fixation to capture, consistent with the current model of genomic looping mediated via protein complexes. Efficient sonication of DNA was not influenced by formaldehyde treatment (Supplementary Fig. 1c).

**CATCH-seq recapitulates ERα ChIA-PET data.** To confirm that CATCH was compatible with next-generation sequencing technology, we chose to target the established enhancer region downstream of the SIAH2 gene. As expected, CATCH followed by next-generation sequencing (CATCH-seq) demonstrated that the oligo-targeted downstream ERE (pull down region) was highly enriched when compared with any other genomic site

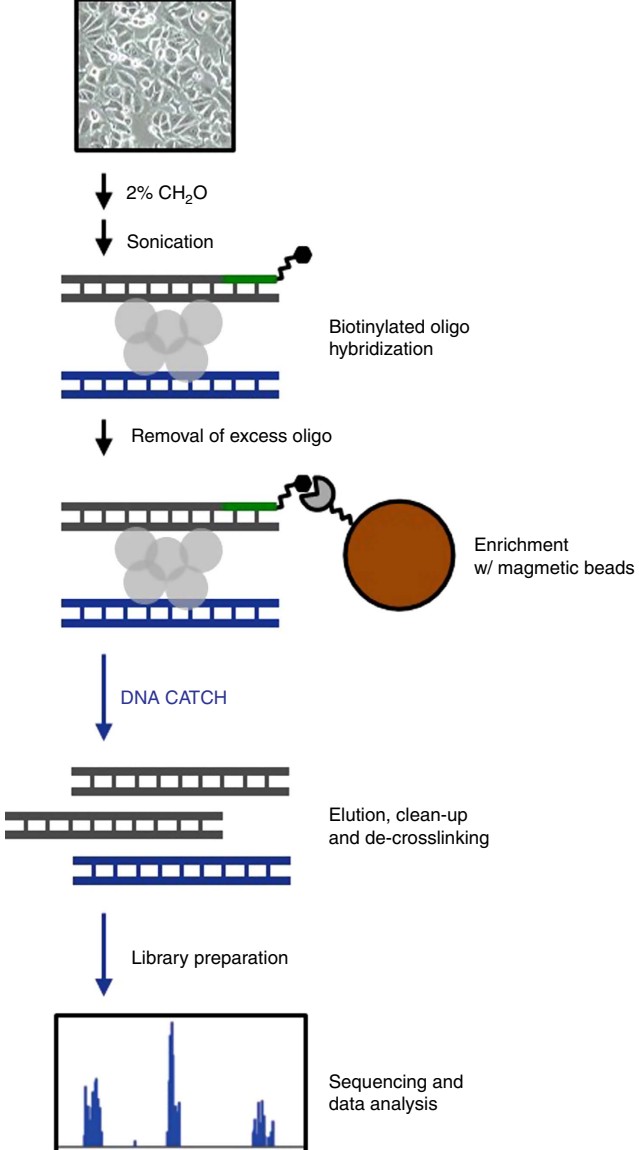

**Figure 1 | The process of CATCH.** The flowchart demonstrates philosophical basis of CATCH. It shows that a cell population is first formaldehyde-fixed to capture DNA–protein–DNA interaction. The DNA is then sheared into small fragments using sonication. The resulting fragmented DNA–protein–DNA complexes are hybridized to a biotinylated oligo in order to enrich for a region of interest. While the targeted sequence is pulled out of the entire DNA population using streptavidin-linked magnetic beads, any associated protein–DNA complexes are also enriched. Subsequent de-crosslinking and PCR amplify the target sequence and any (potentially) associated sequences. Next-generation sequencing allows the identification of any DNA sequences physically associated with the locus (via proteins) to which the biotinylated probe was originally hybridized.

(Fig. 2a). Owing to the nature of the assay, the capture of this region left the DNA of the targeted pull-down locus single stranded, and required second-strand synthesis before sequencing to retain its integrity (Supplementary Fig. 2).

*SIAH2* is an oestradiol-responsive ER target gene with multiple putative EREs located within and adjacent to the gene. In a study focused on identifying functional ER-binding sites, the authors predicted that the intronic region of *SIAH2* contributed to the transcriptional regulation of the gene[18]. However, the promoter region of *SIAH2* does not contain a recognizable ERα-binding

site, and chromatin immunoprecipitation (ChIP) experiments confirmed that ERα binding was nearly undetectable at the promoter, nor was it responsive to E2 (Supplementary Fig. 3). In contrast, both of the tested EREs showed significant increases in ERα occupancy after 45 min of E2 treatment (Supplementary Fig. 3). These data suggested that the intronic and downstream EREs were likely interacting with the *SIAH2* promoter to influence transcription.

In concordance with these data, ERα ChIA-PET analysis of the *SIAH2* gene demonstrated interaction between a portion of the intron and an enhancer region directly downstream of the gene[16]. In addition, the same research group showed interaction between the downstream enhancer of *SIAH2* and multiple other long-distance genomic loci; these data can be visualized through Washington University in St Louis's WashU Epigenome browser (http://epigenomegateway.wustl.edu/). Those data are represented in graphical form (Fig. 2, top).

In order to demonstrate that CATCH-seq could recapitulate data obtained with previously validated techniques, we chose to determine whether it could independently identify ERα-mediated genomic looping interactions attributed to the downstream enhancer of *SIAH2* by Fullwood *et al.* Each of the four long-distance interactions tested were positively demonstrated using CATCH-seq. Interaction with the intronic ERE of *SIAH2* (distance: 17 kb) was demonstrated here in MCF-7 cells, as it was by Fullwood *et al.* (Fig. 2a). Long-distance looping with the ERE upstream of *SIAH2* (distance: 103 kb) was also positively demonstrated, despite its interaction signal appearing ∼2 kb from the previously identified ERα-binding site (Fig. 2b). Interestingly, the downstream ERE of SIAH2 has also been shown to interact with an ERα-binding site (distance: 507 kb) within an enhancer region adjacent to a long non-coding RNA known as *LINC01213*. This interaction was also recapitulated using CATCH-seq (Fig. 2c); however, the interaction signal was lower than that of the other interactions tested. The presence of an interaction within an intron of the *ARHGEF26* gene (distance: 3.4 mb) was also shown to occur as previously identified (Fig. 2d).

For further confirmation, another canonical ERE/promoter interaction on the *TFF1* gene was confirmed (Supplementary Fig. 4), and CATCH was used to demonstrate interaction between progestin response elements and EREs at both the *PDZK1* and *FHL2* genes (Supplementary Fig. 5). In total, these findings indicate that CATCH-seq is capable of reliably reproducing chromatin interaction data that had been previously validated.

***EIF4A1* promoter CATCH-seq.** In order to demonstrate the specificity of CATCH at the level of sequencing, multiple biological replicates were sequenced separately and compared with a 'random' locus capture on the same chromosome (chromosome 17). Previous studies produced data, suggesting that the promoter region of the human *EIF4A1* gene is involved in multiple chromatin interactions with neighbouring loci[19]. In contrast, while a number of interactions occur adjacent to the *GRB7* promoter, it was used as a control pulldown because none of the interactions identified near *GRB7* looked to directly involve the promoter[19]. While the direct capture of both regions of interest was successful (Fig. 3a), the CATCH-seq peaks (representing regions physically interacting with the pulldown region) identified with specific capture of the *EIF4A1* promoter gave highly robust output signal comparative to *GRB7*, hereafter referred to as control (Fig. 3b). Interestingly, many CATCH signal peaks were noted to overlap with histone marks of active enhancers; enhancers were defined as ChIP-seq-positive areas of overlapping H3K27ac and H3Kme1 in T47D cells (Fig. 3b). Analysis of these data for the entirety of chromosome 17 revealed

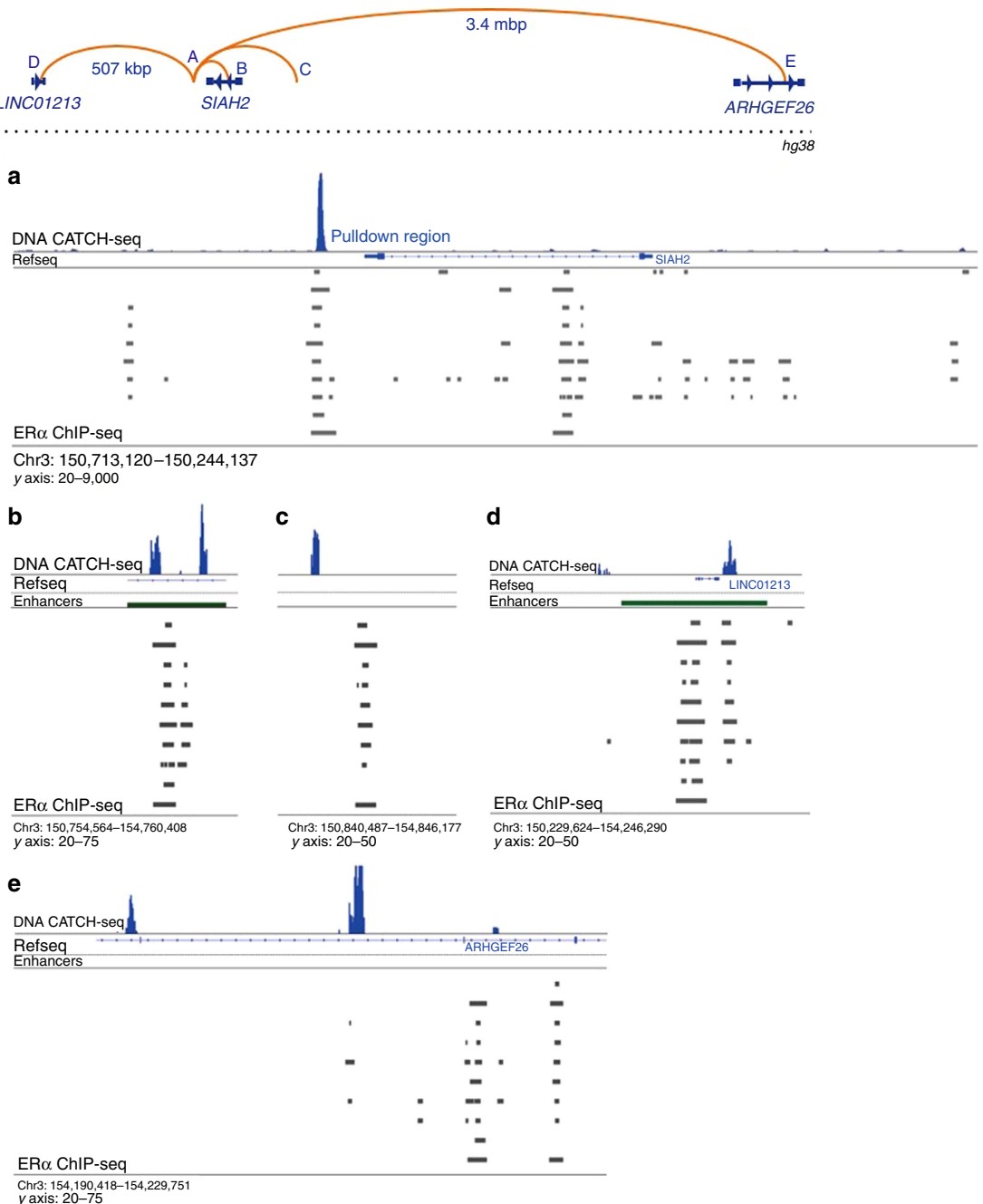

**Figure 2 | CATCH recapitulates chromatin interactions detected with ERα ChIA-PET.** *Top diagram:* the diagram represents the linear distances between (**a**) the downstream enhancer of *SIAH2*, (**b**) a *SIAH2* intronic ERE and (**c–e**) three additional sites shown, using ChIA-PET, to have interaction with site A in MCF-7 cells. The *y* axis labels below integrative genomics viewer (IGV) histograms represent the number of sequencing reads after background subtraction. *Subpanels:* (**a**) the CATCH-seq pull-down region, also an enhancer downstream of *SIAH2*, is demonstrated to be highly enriched after CATCH-seq, confirming the efficacy of the capture method. (**b**) The *SIAH2* intronic ERE, as denoted by the presence of ERα-binding sites detected by 10 different studies (grey bars, data sets outlined in the Methods section), was positive for two different interactions flanking the known ERα-binding regions. (**c**) The downstream ERE shows interaction with an ERα-binding region upstream of *SIAH2*; the location of the interaction was ~2 kb from the ERα-binding site according to CATCH-seq. (**d**) An ERα-binding site near *LINC01213* and (**e**) a region of ERα binding within *ARHGEF26* both show interaction with the *SIAH2* downstream ERE, as previously demonstrated by ref. 16.

that both *EIF4A1* CATCH-seq replicates had ~40% signal overlap with enhancers at lower peak thresholds; however, as the CATCH-seq peak threshold was increased (that is, stronger CATCH-seq signal) this overlap increased to over 80% (Fig. 3c). By contrast, both control capture replicates did not significantly overlap with enhancers above random. These results suggested that enhancers play a significant role in sites of chromatin

looping. By this logic, chromatin looping should contribute to gene expression, thus involving gene promoters. In order to test this hypothesis, CATCH-seq signals within 2,000 bp of a transcription start site (TSS) on chromosome 17 were plotted by density. While the control CATCH-seq produced only random noise, the *EIF4A1* promoter CATCH-seq revealed a striking peak near the TSS of promoters, suggesting a significant DNA–DNA

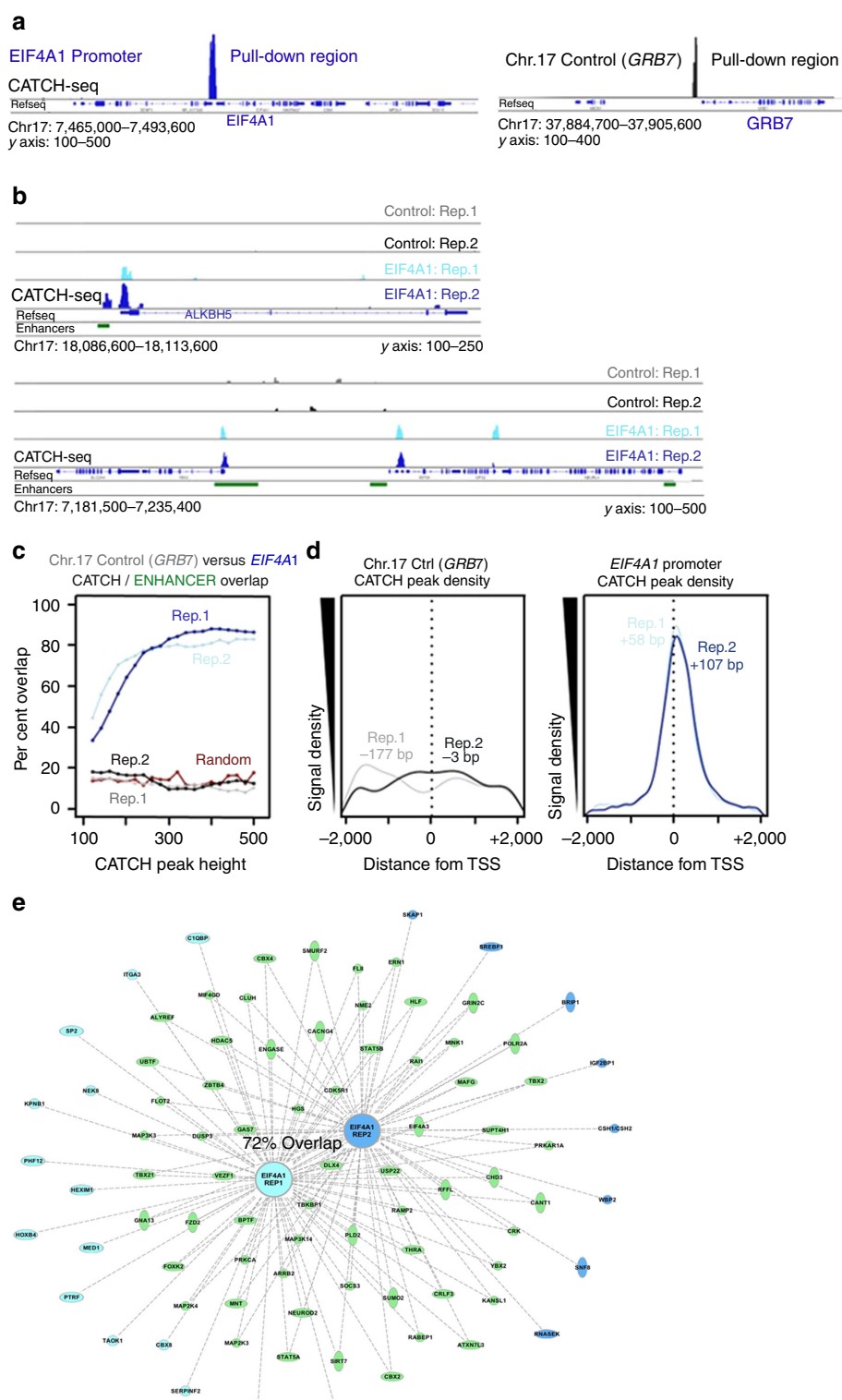

**Figure 3 | CATCH-seq demonstrates specificity and reproducibility.** *y* axis labels below IGV histograms represent number of sequencing reads after background subtraction. (**a**) The promoter region of *EIF4A1* was directly captured via CATCH, along with the promoter region of *GRB7*, which served as a negative control for DNA–DNA interaction. (**b**) Examples of raw CATCH-seq histograms (background-subtracted) demonstrating the locations of peaks near gene promoters, and the reproducibility of multiple replicates. All histograms are on the same scale. (**c**) *EIF4A1* promoter (blue) CATCH-seq interactions significantly overlap with enhancer marks (H3K4me1, H3K27ac) compared with control (greyscale), whose overlap is no greater than statistically random (red); the frequency of these overlaps increase with peak height. (**d**) The CATCH-seq interactions of the *EIF4A1* promoter occur, on average, between 50 and 100 bp downstream of TSSs on chromosome 17. There was no discernable pattern near TSSs for the control pulldown. Both density plots have identical *x* and *y* axes. (**e**) Ingenuity pathway analysis (IPA) analysis detailing specific gene promoters that demonstrated physical interaction with the *EIF4A1* promoter. There was a remarkable 72% overlap between the CATCH-seq replicates (shared interactions: green, unique interactions: light and dark blue).

interaction enrichment (Fig. 3d). In addition, when the replicates were evaluated via Ingenuity Pathway Analysis (IPA), the two *EIF4A1* CATCH-seq replicates had 72% overlap in identified signal peaks, and a similarly high degree of overlap in identified gene promoters, suggesting an exceptionally high degree of reproducibility among biological replicates for sequencing experiments (Fig. 3e).

To determine whether these observations held true on a different chromosome, an enhancer downstream of the *MYC* gene was interrogated in a similar manner (Supplementary Fig. 6a). As previously, a large proportion of CATCH-seq signal resulting from the *MYC*-downstream enhancer capture overlapped with enhancer marks, and this proportion trended exponentially upwards with signal strength, growing to over 80% at higher peak thresholds (Supplementary Fig. 6b). CATCH-seq peaks within 2,000 bp of a TSS were also centred near the TSSs on chromosome 8, again falling ~112 bp downstream of the TSS, reinforcing the data from Fig. 3d (Supplementary Fig. 6c). These data again suggested that enhancers play a critical role in DNA–DNA interactions that occur near gene promoters, as can be seen in this interaction between the captured enhancer and the *AZIN1* promoter (Supplementary Fig. 6d, top). Critically, while the CATCH-seq data demonstrated that the captured enhancer downstream of *MYC* was interacting with more than a single promoter, not every gene promoter harbours an interaction signal, suggesting that CATCH-seq is capturing authentic DNA–DNA interaction. One example is the *E2F5* promoter, which demonstrates strong interaction with the enhancer downstream of *MYC* at the exclusion of other genes in the area (Supplementary Fig. 6d, bottom). IPA found the processes of gastric carcinoma and homologous recombination to be the top pathways regulated by this enhancer (Supplementary Fig. 6e). In short, these data suggest that the captured enhancer downstream of *MYC* is capable of physically interacting with a host of additional enhancers and gene promoters across chromosome 8, and that the genes involved in these interactions can be linked to common disease/biochemical processes.

**ER activation alters downstream chromatin interactions**. It has been debated whether genomic organization is largely plastic, static, or somewhere between those two extremes. Notably, research has demonstrated both plastic and static types of gene promoters[20], and genome-wide analyses suggest a non-zero probability that any two parts of the genome, no matter how distant, to be interacting, suggesting a high degree of plasticity[21]. Oestradiol is a powerful genome-wide transcriptional inducer via ER activation, and, as such, was used to determine whether the chromatin interactions of enhancers could demonstrate such plasticity. Because *SIAH2* transcription is upregulated upon ER activation, its downstream enhancer was an optimal target for such experiments.

Despite *SIAH2* transcription being activated by oestradiol treatment, both vehicle and oestradiol treatments showed DNA–DNA interaction between the downstream enhancer (pull-down region) and the *SIAH2* intron (as demonstrated by Fullwood *et al.* previously) and promoter (Fig. 4a). As with the other CATCH-seq pulldowns on chromosomes 8 and 17, the CATCH-seq signal density within 2,000 bp of a TSS peaked significantly (compared with randomized distribution of the data set) near the TSSs of chromosome 3, averaging ~200 bp downstream for both vehicle and oestradiol treatments (Fig. 4b). Strikingly, when comparing genes on chromosome 3 whose promoters were identified as interacting with the downstream enhancer of *SIAH2*, vehicle and oestradiol treatments shared 52% overlap (264 genes), suggesting that oestradiol treatment was not responsible for completely

rearranging genomic structure (Fig. 4c). However, oestradiol treatment lost 107 enhancer–promoter interactions from vehicle baseline, while it gained 140, demonstrating a critical plasticity in genomic architecture that could potentially play a role in altering transcriptional programmes and activity. Many of these types of interactions are illustrated in Fig. 4d, showing individual examples of enhancer–promoter interactions being gained upon oestradiol treatment, as well as significant overlaps with CATCH-seq signals and enhancer sites on chromosome 3 (Fig. 4d). Similar to previous analyses, a large proportion of the CATCH-seq signal resulting from the *SIAH2* downstream enhancer capture overlapped with enhancer marks. Interestingly, a similar trend was observed for overlap with locations of ER binding in both the presence (Fig. 4e, left) and absence (Supplementary Fig. 7) of oestradiol, and at higher CATCH-seq signal thresholds this overlap reached well over 90%. While ER-binding sites were highly correlated to sites of chromatin looping for this particular enhancer in both the presence and absence of oestradiol, it was clear that oestradiol had a significant impact on chromatin architecture. Not only did oestradiol trigger a chromosome-wide alteration in the genes interacting with the downstream enhancer of *SIAH2*, but the total number of genes identified by CATCH-seq was higher (at all thresholds) in the presence of oestradiol (Fig. 4e, right). These results argued a great deal of plasticity in DNA–DNA interactions in response to oestradiol. This was further illustrated via IPA, which elucidated highly distinct subsets of genes in the presence and absence of oestradiol (Fig. 4f). Interestingly, the RNA expression process was significantly enriched upon oestradiol treatment (Fig. 4f, blue), whereas there was no enrichment of this pathway in the absence of oestradiol (Fig. 4f, grey); these findings support the function of oestradiol as a genome-wide gene expression modulator. Together, these findings suggested the potential for malleable transcriptional foci, containing the interactions of many genes at once; many enhancer–enhancer or enhancer–promoter interactions within such a hub would be stable, but some could form or dissipate based on changing biochemical stimuli.

**CATCH-seq predicts correlation with gene expression**. In order to test the hypothesis that a single enhancer is capable of interacting with, and altering the transcription of, multiple gene targets on the same chromosome, the SEEK algorithm (search-based exploration of expression compendia; http://seek. princeton.edu/) was employed. SEEK determines gene expression correlation by weighting available gene expression data sets based on input genes of interest[17]; using this weighted correlation aggregation method, it calculates relative gene coexpression among those data sets. If CATCH-seq was truly identifying DNA–DNA interactions at gene promoters that led to the alteration of transcriptional expression of that gene, CATCH-seq data should be significantly more proficient at predicting coexpression of gene cohorts than random. If the list of gene promoters identified via CATCH-seq is significantly enriched (over random) for genes that are also transcriptionally coexpressed, it would suggest that the long-distance genomic interactions of a single enhancer are capable of influencing gene expression patterns, not just the transcriptional output of a single gene.

To test this hypothesis, the top 500 gene promoters for each CATCH-seq experiment were identified. Next, SEEK lists were created using a 'seed list'; a three-gene subset (described in detail within the Methods) specific to each CATCH-seq experiment; each SEEK list was then sorted based on the highest coexpression value. The top 100 coexpressed genes from the pull-down chromosome of interest for each CATCH-seq experiment were

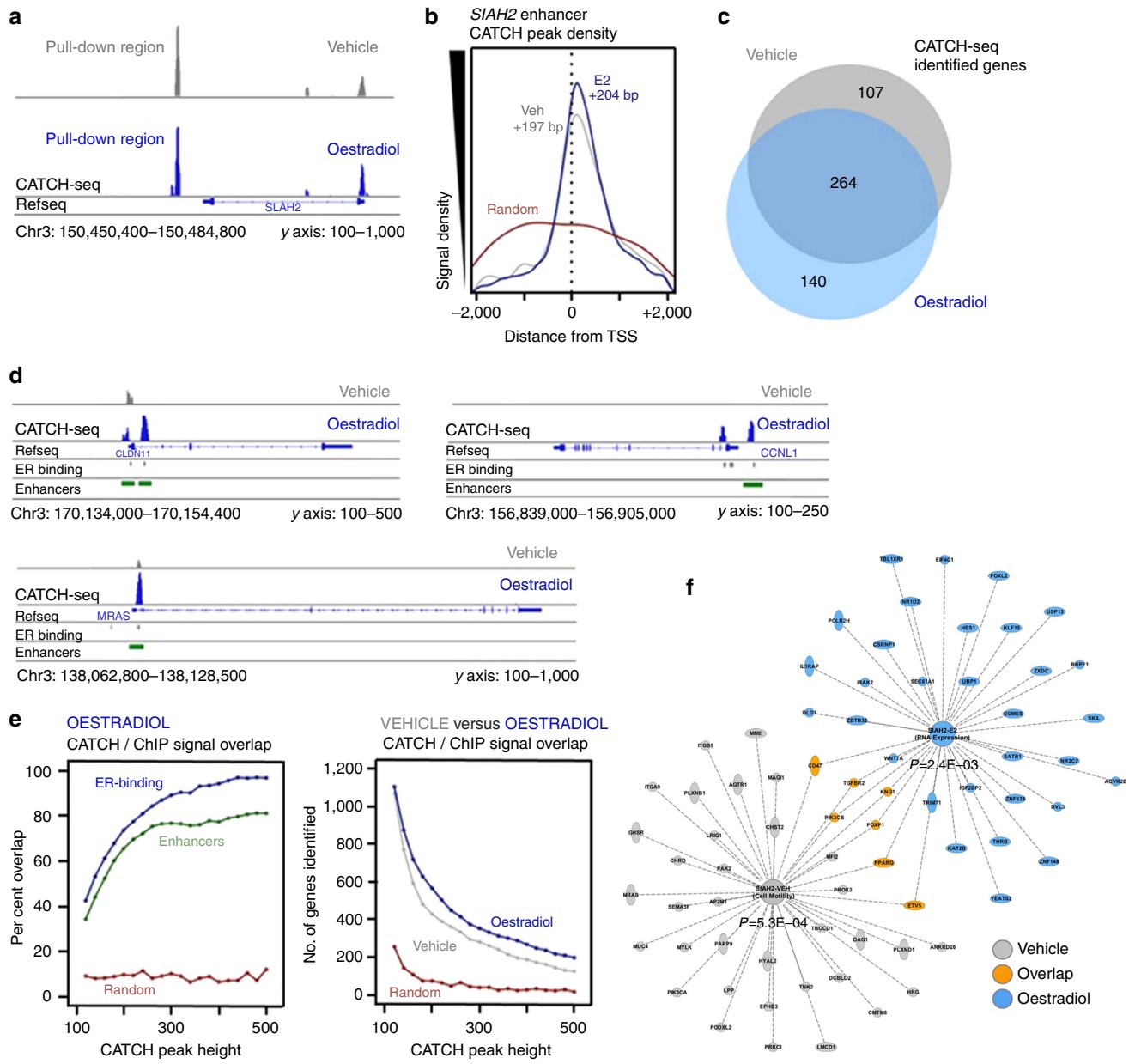

**Figure 4 | CATCH-seq of enhancer downstream of SIAH2 reveals plasticity of genomic architecture.** *y* axis labels below IGV histograms represent number of sequencing reads after background subtraction. (**a**) The captured region is the downstream enhancer of *SIAH2* (T47D human breast cancer cells). Interaction peaks can be seen at the intronic enhancer and *SIAH2* promoter in both the presence and absence of oestradiol. (**b**) The CATCH-seq signal density within 2 kb of TSSs on chromosome 3 peaks at around +200 bp in both the presence (blue) and absence (grey) of oestradiol. There was no discernable peak in the randomized data set (red). (**c**) To-scale Venn diagram demonstrating the number of genes' promoters identified as interacting with the downstream enhancer of SIAH2. The majority of genes are unchanged with the addition of oestradiol (264); however, oestradiol treatment does induce the loss of interaction (107) and the gain of interaction (140) of a large subset of genes. (**d**) Examples of raw CATCH-seq histograms demonstrating the striking overlap between ERα-binding sites, enhancer marks and CATCH-seq peaks near the promoters of genes. Many of the peaks (which indicate physical interaction with the downstream enhancer of *SIAH2*) are reduced/not found in the absence of oestradiol, or enhanced upon the addition of oestradiol. (**e**, left) *SIAH2* enhancer CATCH-seq interaction peaks significantly overlap with enhancer marks (green) and ER-binding sites (blue) compared with statistically random control (red); the frequency of these overlaps increases with peak height. (**e**, right) As expected, increasing peak height thresholds reduce the number of CATCH-seq peaks identified at gene promoters; however, at all thresholds, oestradiol treatment facilitates more DNA–DNA interactions than vehicle control. (**f**) IPA analysis of two statistically significant processes identified under vehicle- (grey; cell motility) and oestradiol-treated (blue; RNA expression) conditions. There is little overlap (orange) between the two processes.

denoted by the SEEK list. A flowchart describing the processing of each data set can be found in Fig. 5a. The *EIF4A1* promoter CATCH-seq identified 16 genes also present on the SEEK list; by contrast, a random SEEK list created from only genes found on that chromosome (chromosome 17) could only identify an

average of nine genes in common with the CATCH-seq experiment (Fig. 5b, right). Not surprisingly, the control capture on chromosome 17 was incapable of identifying any genes on the SEEK list (Fig. 5b, left). Strikingly, capture of the enhancer downstream of *MYC* yielded an overlap of 46-out-of-100 genes

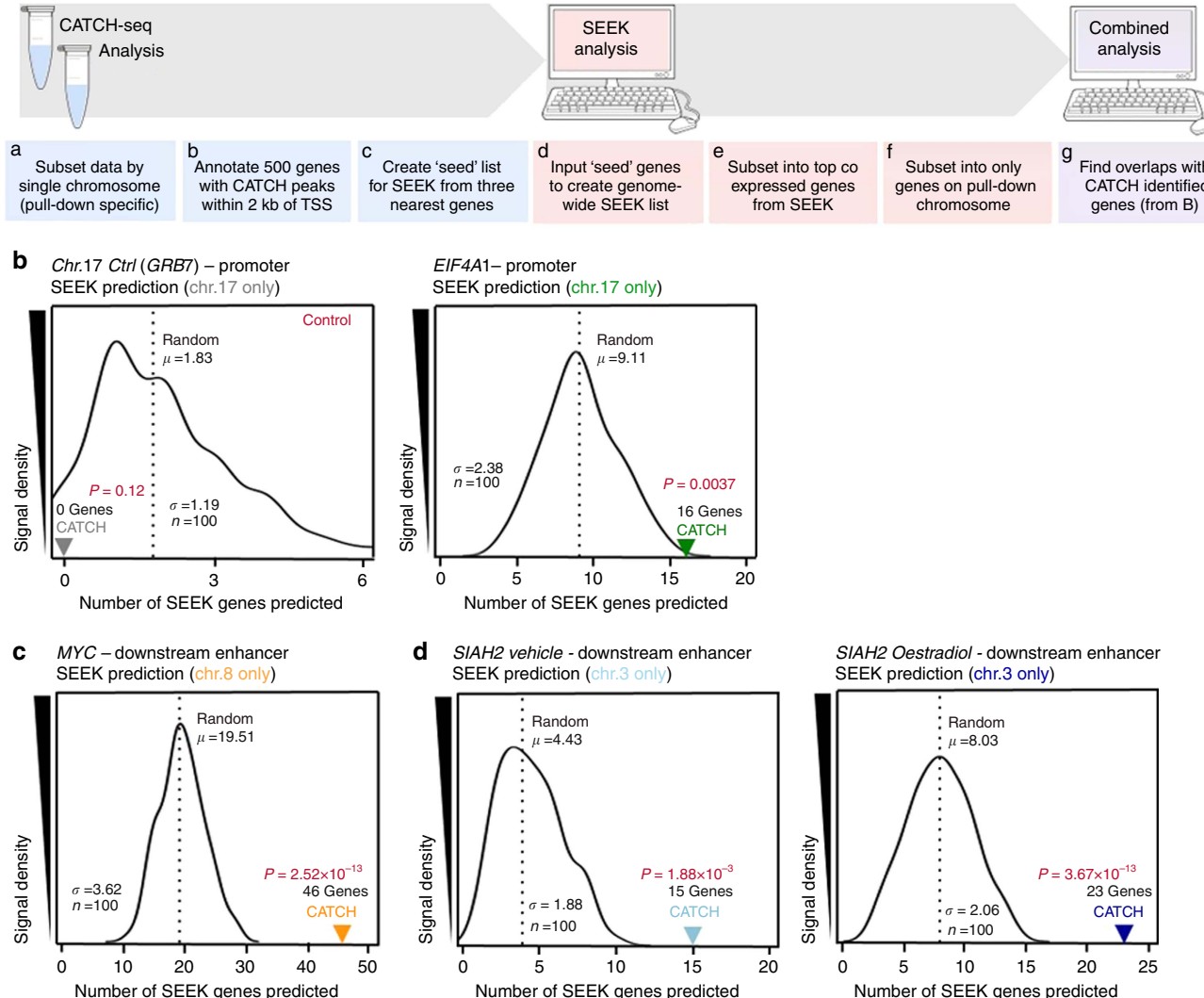

**Figure 5 | Finding CATCH-seq interactions at gene promoters predicts transcriptional coexpression.** (**a**) Graphic outlining the general processing and data analysis that assess the ability of CATCH interactions to predict SEEK coexpressed genes. (**b**) Histogram representing the number of SEEK genes 'predicted' at random, or by specific CATCH-seq (left, grey: *GRB7* negative control; right, green: *EIF4A1* promoter). (**c**) Histogram representing the number of SEEK genes 'predicted' at random, or by specific CATCH-seq (orange: *MYC* downstream enhancer). (**d**) Histogram representing the number of SEEK genes 'predicted' at random, or by specific CATCH-seq (left, light blue: *SIAH2*–vehicle; right, dark blue: *SIAH2*–oestradiol). In each experimental case, CATCH was capable of significantly predicting SEEK coexpression over random. All analyses were restricted to the specific chromosome of the CATCH pulldown. *P* values were calculated via *t*-distribution.

from its unique SEEK coexpression list (Fig. 5c). Similarly, both conditions of *SIAH2* downstream enhancer CATCH were able to predict significantly more genes than would be expected at random (Fig. 5d), as the CATCH prediction results were statistically outside the entire histogram of random prediction. These results demonstrated that CATCH-seq was capable of identifying DNA–DNA interactions involving subsets of gene promoters that are transcriptionally linked (coexpressed). In order to show that this predictive power was not a function of using too selective SEEK gene list, additional SEEK genes (100, 200 and 500) were used in the CATCH/SEEK analysis. In all cases, CATCH was still able to predict significantly more SEEK genes than random (Supplementary Table 4).

The presence of oestradiol was able to induce changes in the DNA–DNA interactions of the downstream enhancer of *SIAH2* (Fig. 4). In order to determine whether CATCH-seq interactions were able to predict gene coexpression under these conditions,

unique 150 gene SEEK lists were created for the vehicle- and oestradiol-treated samples of the downstream enhancer of *SIAH2* CATCH-seq experiment. The majority of genes predicted to be coexpressed were identified in both the vehicle- (93%) and oestradiol-treated (74%) conditions (Fig. 5c, orange). However, consistent with the data that demonstrated the presence of oestradiol increased the total number of chromatin interactions (Fig. 4e), oestradiol treatment also resulted in more CATCH-seq-predictive power. In total, the oestradiol-treated condition of the downstream enhancer of *SIAH2* CATCH-seq was able to predict the coexpression of 23 genes (Fig. 5c, blue), compared with an average of 0.64 genes predicted at random, while the vehicle-treated condition of the same experiment predicted the coexpression of only 15 genes (Fig. 5c, grey). Interestingly, *SIAH2* was identified as one of the top coexpressed genes upon oestradiol treatment, but not vehicle (Supplementary Table 5). Because the captured region in this particular CATCH-seq

experiment is a known enhancer of *SIAH2*, these results strongly suggested that the CATCH-seq interactions were making biologically meaningful predictions of transcriptional coexpression. IPA was also performed on these highly limited subsets of 15 and 23 genes for vehicle and oestradiol treatments, respectively (Supplementary Fig. 8a). Interestingly, both subsets of genes were significantly linked to female-specific cancers (Supplementary Fig. 8), supporting the ideas that the downstream enhancer of *SIAH2* is both hormone-responsive, and is capable of influencing the expression of a host of genes involved in these processes. Despite the majority of 'predicted' genes being common to both vehicle and oestradiol treatments, the most significant enriched pathways were defined largely by unique genes (Supplementary Fig. 8b). Together, these data strongly support the concept that single enhancers are capable of regulating a host of transcriptionally coexpressed genes, even at linear distances of multiple millions of base pairs.

## Discussion

The importance of genome-wide DNA interactions that coordinate the functional regulation of virtually every gene in the human body is beginning to gain recognition[22]. Unfortunately, the current suite of assays available to probe DNA–DNA interaction are time-consuming, difficult to use and come with inherent flaws and difficulties that limit their practical effectiveness. A more complex adaptation of 3C technology (5C) solved some issues with high-volume experimentation, but could not be applied to genome-wide interrogation because of the number of primer sets required[23]. More recently, Hi-C technology was developed in attempt to take the 3C line of technologies genome-wide[24]. Unfortunately, despite the use of tagged nucleotides for more reliable data capture, Hi-C technology requires, like its predecessors, enzymatic digestion and ligation. It has been shown that when enzymatic digestion and ligation are used, less than 1% of DNA fragments actually yield ligation products, suggesting massive data loss[12]. The recently devised Targeted Chromatin Capture (T2C) method, which also uses the problematic enzyme-based digestion and ligation, has been able to increase the resolution of chromosomal capture[25]. However, T2C requires the hundreds of custom-made oligonucleotides that have a strict list of guidelines for design, and necessitates the building of a custom oligo array. Such processes require excessive time and funding for the average research laboratory.

Here we have devised a simplified method that offers significantly increased resolution for the detection and characterization of chromatin looping and long-distance genomic communication associated with a single locus. One limitation of this technology is the necessity of choosing the locus of interest, such as a promoter or known enhancer. Offsetting the targeted nature of the assay are a number of significant advantages: CATCH is highly reproducible, does not require enzymatic digestion or ligation and can be completed in less than 24 h—all done with unparalleled bp resolution. It is also important to emphasize that this technology, while limited to the capture of a single locus, can still assess genome-wide relationships when paired with next-generation sequencing. The current study has employed the technology only over the span of single chromosomes because of a limited number of sequencing reads available in the experiments. With deeper sequencing coverage, genome-wide CATCH-seq data would easily be available and significant.

Because CATCH requires so little time to complete, the assay has been streamlined at every step. As with any biological assessment, there is inherent variability as well as critical steps to improve functionality. As expected, efficient sonication was critical to abolish false-positive signals, while oversonication resulted in loss of signal; these findings are similar to ChIP protocols, where efficient sonication is arguably the most critical step to obtaining high-quality data[26]. In addition, because each of the interacting loci is directly captured through chemical crosslinking, there is no lost data because of improper DNA ligation or unexpected ligation products. These improvements allow CATCH to capture DNA interactions that were previously impossible to detect, and do so in a reduced number of cells. This suggests that CATCH could be used in studies where limited material is present, such as those utilizing human tissue samples.

The design of the biotinylated capture oligonucleotide was also a critical, but very simple, step to ensure maximum assay efficiency. The biotin was attached at the 5′ end of the oligonucleotide using a 15-atom triethylene glycol (TEG) spacer that eliminated steric hindrance between the biotin moiety and the target DNA–protein complexes, allowing full accessibility to the streptavidin magnetic beads. It was also determined that a desthiobiotin moiety could be used in place of biotin, allowing for a gentler elution from the beads with the addition of excess biotin as shown previously[27].

In this study, we demonstrated that CATCH could not only recapitulate previous findings, but was also able to detect previously unreported long-distance chromatin interactions. Earlier ChIP-3C studies demonstrated an interaction between the intronic ERE[B] and the ERE downstream of the *SIAH2* gene[16] and our work confirms this interaction. However, CATCH-seq also demonstrated the existence of a highly complex web of interactions between the downstream enhancer of *SIAH2* and multiple enhancers and promoters spanning the entirety of chromosome 3; this finding also held true for loci on chromosomes 8 and 17. While the data presented here support the idea that gene promoters are being physically linked, the biochemical data paint a slightly different picture. Traditionally, gene promoters are thought to span ~5 kbp upstream of a genes TSS, however our CATCH-seq data suggest that the average chromatin-looping interaction involving the TSS region of genes occurs between 50 and 200 bp downstream of the TSS. These findings could be significant for future research projects interested in the specific nature of chromatin architecture and the potential DNA motifs that may coordinate such interactions.

The idea that multiple promoters could be involved in a single 'hub' of chromatin interaction is not novel, but has little experimental evidence to date. The data presented here strongly support the hypothesis that single enhancers can regulate a host of genes, even at linear distances of multiple millions of base pairs. While the exact mechanism by which these transcriptional hubs may function is yet unknown, our data demonstrate that subsets of genes, spanning entire chromosomes, can physically associate with the same enhancer, and that a highly significant portion of those genes are coexpressed within the cell. In relation to these transcriptionally associated DNA–DNA interactions, the CTCF protein has been implicated in mediating such looping[28]. For this reason, we assessed to what degree CATCH peaks and CTCF-binding sites were adjacent or overlapping in T47D cells. Despite it's documented role in looping, there was very little overlap found between CTCF-binding sites and CATCH peaks (Supplementary Fig. 9). This could be explained by the fact that CTCF need not bind directly to these sites in order to influence the looping of such sites, or that it is not needed at all, as cohesin can mediate such interactions and binds to enhancers independently of CTCF[29]. Despite all these data, our findings do not directly relate DNA–DNA interaction to active, ongoing transcription; the correlation of CATCH and SEEK suggests a relationship between these interactions and general transcriptional networks, but cannot speak to the nature of these interactions in any given single cell.

The limitations of the CATCH assay are apparent, not from a base-pairs-of-resolution standpoint like ChIP, but seem to be more attached to the number of CATCH interactions needed to begin to statistically identify subsets of interacting genes. This point is illustrated in Supplementary Fig. 10, where, for various depths of SEEK list genes (100, 150, 200 and 300), the negative $\log_{10}$ of the CATCH $P$ value (for SEEK gene prediction) was plotted on the $y$ axis against the number of CATCH genes identified in the analysis on the $x$ axis. Several interesting points arose from these analyses. First, the approximate resolution of CATCH's transcriptional coexpression-predictive power (as measured by SEEK list prediction) increases with decreasing SEEK gene depth. This effectively suggests that CATCH is more reliable at predicting the most significant SEEK genes, as might be expected with such complex biology. Second, while each CATCH experiment has slightly different effective resolution, they all trend similarly; on average, the effective resolution of CATCH falls between identifying ∼100–200 genes. This, to a degree, attempts to measure the resolution of the assay, not in base-pair like ChIP, but in number of identified genes, making CATCH a technique that functions very well in predicting large cohorts of transcriptionally coexpressed genes, but less reliably for single locus–locus contacts. Attempting to identify fewer genes than the effective resolution of CATCH may not result in the identification of statistically significant coexpressed genes, as it may be influenced by noise and the variations inherent to the complex nature of this biology. Interpretations of single-gene locus–probe interaction should be used with caution. Nonetheless, by summarizing interactions to hundreds of identified genes, this analysis illustrates the collective effect of interactions in mediating gene coexpression.

In summary, these studies describe and validate a novel next-generation technique for the detection of DNA–DNA interaction called CATCH. Importantly, this technique has demonstrated the ability to detect previously undetectable long-distance chromatin interactions, suggesting that genome-quaternary structure may be much more complex than initially believed, and that it could play a significant role in the expression of entire gene programmes. This revelation would be highly significant in the areas of epigenetics, disease therapy and genome biology.

## Methods

**CATCH.** This is an original protocol. What follows is an abbreviated version; the full detailed user version is available below, including all buffer recipes. MCF-7 or T-47D cells (log-phase growth, ∼1 × 10⁶ cells per sample) were fixed for 6 min at room temperature with a final concentration of 1% formaldehyde (fresh single-use vials) at ∼50% culture confluency. Crosslinking was quenched for 10 min at room temperature by the addition of Tris-HCl pH 8.0 to a final concentration of ∼0.125 M. Next, the cells were harvested via scraping in 1 ml of PBS into a 1.5 ml Eppendorf tube, and then spun at 250g for 8 min (all centrifugation steps were carried out in a standard tabletop microfuge). The supernatant was aspirated and the cells were resuspended in 500 μl of cold Nuclear Isolation Buffer supplemented with a protease inhibitor cocktail (Calbiochem). The samples were dounce-homogenized 20 times with a tight-fitting pestle and centrifuged again for 10 min at 750g to pellet nuclei. The supernatant was aspirated and the nuclear pellets were resuspended in 100 μl of CATCH buffer supplemented with protease inhibitor cocktail. The samples were sonicated for two cycles (HIGH, 30 s on/off) of 8 min each in a Diagenode BioRuptor sonication device, and the cellular debris was pelleted by centrifugation at 24,000g for 15 min at 4 °C. Sonication efficiency was then assessed on a 1.5% (w/v) agarose gel. Genomic DNA fragments should largely fall between 100 and 500 bp. Next, the sheared chromatin sample was incubated at 58 °C for 5 min to unmask biotin on endogenous proteins. Then, 10 μl of pre-equilibrated (in CATCH buffer) streptavidin magnetic beads (Thermo Scientific) were added to each sample. The samples were incubated for 1 h at room temperature while gently rotating. Next, the magnetic beads were extracted and the supernatant was transferred to a clean PCR tube. To each sample, specific bioti-nylated oligonucleotide probe (Integrated DNA Technologies) was added to a final concentration of ∼300 nM. The probe was then hybridized by incubating the samples as follows: 25 °C for 2 min, 81 °C for 4 min (denaturation), 72–42 °C decreasing gradient (12 s per degree), 42 °C for 30 min (hybridization), followed by storage of the sample at 25 °C. It is important to note that, during testing,

denaturation temperatures below 75 °C or above 85 °C were detrimental to oligonucleotide annealing or long-range interaction detection, respectfully; impact on interaction detection at 81 °C was undetectable. The hybridized sample was then transferred to a new 1.5 ml Eppendorf tube and any unhybridized biotinylated oligo was removed with an Illustra Sephacryl (S-400HR) spin column, according to the manufacturer's instruction. The cleared product was again transferred to a new 1.5 ml Eppendorf containing 300 μl of nuclease-free H₂O. Next, 25 μl of pre-equilibrated (in CATCH buffer) streptavidin magnetic beads were added to each sample. The samples were incubated at room temperature for 1 h while gently rotating. The beads from each sample were then immobilized on a magnetic stand and washed five times in CATCH buffer at 42 °C while shaking at 1,000 r.p.m. in a thermomixer. The beads were then resuspended in 150 μl of de-crosslinking buffer supplemented with 5 μl of 20 mg ml⁻¹ Proteinase K. The sample was incubated at 55 °C for 30 min to while shaking at 1,000 r.p.m. on a thermomixer, followed by incubation at 65 °C overnight on the same thermomixer. Finally, the sample was incubated at 100 °C for 60 s to destroy any remaining biotin–streptavidin binding and elute the DNA from the magnetic beads. The supernatant was immediately transferred to a new 1.5 ml Eppendorf tube. The DNA was then purified using phenol–chloroform–isoamyl alcohol, and then precipitated in 100% ethanol using glycogen (Thermo Scientific) as a carrier. The DNA was pelleted by spinning at 24,000g for 25 min at room temperature, and resuspended in TE buffer. A complete user protocol is available in the Supplementary Materials. An overview comparison of CATCH and other chromosome capture methods is available in Supplementary Table 1.

**CATCH-biotinylated oligo design.** Biotinylated oligonucleotides were ordered from Integrated DNA Technologies, using the TEG–biotin modification on the 5′ end of the oligo. All oligos were designed with Primer3 version 4.0 to be between 23 and 25 nucleotides in length, with a $T_m$ as close to 63 °C as possible. Testing multiple oligonucleotides, it was found that those biotinylated oligos targeted to regions ∼150 bp from the targeted protein-binding site gave the most reliable data. The oligos were resuspended at 1 μg μl⁻¹ in TE buffer and stored at −20 °C until use.

**Cell culture.** MCF-7 (American Type Culture Collection (ATCC); HTB-22) and T-47D (ATCC; HTB-133) cells were maintained in phenol red-free RPMI 1640 with L-glutamine supplemented with 10% (v/v) heat-inactivated fetal bovine serum and 100 U ml⁻¹ penicillin–streptomycin. Cells were housed at 37 °C in 5% CO₂ for a maximum of 12 passages after being purchased directly from the ATCC. The cells used in Fig. 2 for the validation of CATCH-seq (when compared with ChIA-PET) were MCF-7 at passage 3 after being purchased directly from the ATCC (HTB-22). According to the ATCC (MCF-7), the cytogenetic analysis yielded a modal chromosome number of 82, with a range of 66–87. The stemline chromosome numbers ranged from hypertriploidy to hypotetraploidy, with the 2S component occurring at 1%. There were 29–34 marker chromosomes per S metaphase; 24–28 markers occurred in at least 30% of cells, and generally one large submetacentric (M1) and three large subtelocentric (M2, M3 and M4) markers were recognizable in over 80% of metaphases. No double minutes (DM) were detected. Chromosome 20 was nullisomic and X was disomic.

**CLOVER analysis.** Using the freely available CLOVER (http://zlab.bu.edu/clover/) programme[30] according to the specified instructions, full-site oestrogen response elements were identified within and around the SIAH2 gene ±100 kb. The resulting potential binding sites were then cross-referenced to previously identified ER-binding sites within MCF-7 cells[31]. For this analysis, the NCBI36/hg18 build of the human genome was used. Full-site EREs identified by CLOVER and/or positively correlated with previous data were then used in the subsequent ChIP and CATCH assays. These sites are detailed in Supplementary Fig. 3.

**ChIP analysis.** Cells were fixed with 1% formaldehyde for 10 min at room temperature. Reaction was quenched with glycine, and cells were centrifuged to pellet and resuspended in ChIP lysis buffer (10 mM Tris pH 8.0, 10 mM NaCl, 5 mM EDTA, 1% NP-40, 1% SDS, 0.5% deoxycholate) supplemented with protease inhibitors. The cell slurry was incubated for 10 min on ice and then sonicated for three cycles (HIGH, 30 s on/off) of 7 min each in a Diagenode BioRuptor soni-cation device, and the cellular debris was pelleted by centrifugation at 24,000g for 15 min at 4 °C. Sonication efficiency was then assessed on a 1.5% (w/v) agarose gel. Genomic DNA fragments largely fell between 100 and 700 bp. Next, the sheared chromatin sample was diluted to 1 ml in ChIP Dilution Buffer (17 mM Tris pH 8.0, 33 mM NaCl, 1% SDS, 0.5% NP-40) supplemented with protease inhibitors. Here 10% of total volume was taken as input. Then, 2 μg of anti-ERα (Santa Cruz Biotechnology, HC-20) or rabbit IgG antibody was added to each sample, and the samples were rotated overnight at 4 °C. Next, magnetic protein-G Dynabeads (Invitrogen) were washed once in PBS supplemented with 5% BSA and resus-pended in ChIP dilution buffer. Then, 30 μl of the pre-washed beads were added to each sample, and the samples were rotated at 4 °C for 2 h. The beads were then washed consecutively in ChIP Wash Buffer I (20 mM Tris pH 8.0, 150 mM NaCl, 2 mM EDTA, 1% NP-40, 1% SDS), ChIP Wash Buffer II (20 mM Tris pH 8.0, 500 mM NaCl, 2 mM EDTA, 1% NP-40, 1% SDS), ChIP Wash Buffer III (20 mM

Tris pH 8.0, 250 mM LiCl, 1 mM EDTA, 1% NP-40, 1% Deoxycholate) and TE buffer. The beads were then resuspended in 100 μl freshly made ChIP elution buffer (200 μl of 10% SDS and 0.168 g of NaHCO₃ in 2 ml of H₂O). Next, the samples were incubated at 65 °C for 15 min to elute the complex from the beads. That process was repeated and the eluates were combined. Finally, 8 μl of 5.0 M NaCl was added to each sample (including input samples) and they were incubated overnight at 65 °C. Samples were incubated with RNase and Proteinase K before processing with the QIAquick PCR purification kit (Qiagen) according to the manufacturer's instructions. PCR primers for individual ChIP experiments are detailed in Supplementary Figs 4 and 5. ChIP for ERα (Santa Cruz Biotechnology, sc-542) and H3K4me1 (Millipore, 07–436)/H3K27ac (Millipore, 07–360; Figs 3 and 4) were performed as above and followed by Illumina next-generation sequencing at the UChicago Sequencing Facility. For each ChIP, 2 μg of antibody was used per experiment.

**PCR quantification (ImageJ).** For Supplementary Figs 4 and 5, PCR products were diluted in 6× Orange G loading buffer and run at 100 V for 28 min on a 1.5% agarose gel with ethidium bromide (ladder was Bioline EasyLadder I). The resulting gel was imaged under ultraviolet light, and the individual bands were quantified via ImageJ using the measure function. First, the background value for each band was taken. Next, the value of the band itself was taken, and the background value was subtracted from the value of the band. Each resulting value was then normalized to the value of the targeted pulldown in the experiment, such that the value of the pulldown became 1.0. This ensured subtraction of background variation and random variation in pull-down efficiency. The resulting values were then plotted as the mean with error bars of s.e.m.

**Creation of sequencing libraries.** DNA (10 μl) from CATCH final elution was immediately (without freezing) put through the second-strand synthesis protocol using NEBNext Module #E6111S according to the manufacturer's instructions. Nucleic-acid-binding beads were AMPure XP #A63881, purchased from Agencourt. Next, the DNA template was made into a sequencing library using the KAPA Biosystems library kit #KK8232 following the manufacturer's instruction. The KAPA kit was critical as it produces a library with fewer 'bead swap' steps, allowing you to retain a better DNA template yield, and thus makes a library from less starting material. A complete user protocol is available in the Supplementary Materials; sequencing depth for each library varied between ~15 and ~24 million reads: GRB7 replicates 1 and 2 had 16.0 and 22.9 million reads, respectively; MYC had 16.1 million reads; EIF4A1 replicates 1 and 2 had 18.6 and 24.2 million reads, respectively; SIAH2 vehicle-treated had 16.2 million reads; and SIAH2 oestradiol-treated had 15.1 million reads.

**Creation of SEEK lists.** The SEEK algorithm can be found here http://seek.-princeton.edu/, and is a web application stemming from research carried out at the Princeton University[17]. SEEK allows a number of genes as input to determine a ranked-order list by coexpression. This coexpression rank is a comprehensive analysis based on over 5,000 independent microarray and RNA-sequencing data sets. In T-47D cells, to create each SEEK list, three 'seed' input genes were selected with the following rules: (a) the gene must have a CATCH-seq peak within 2 kb of its TSS and the peak must be above background, (b) the peak near the TSS of the gene must be one of the top 500 (in height) such peaks on the chromosome and (c) the gene must not be the primary target of the enhancer, according to our study (for example, SIAH2 was not used as a 'seed' gene, despite being identified in the associated CATCH experiment, wherein the downstream enhancer of SIAH2 was captured). Using those guidelines, the three gene promoters nearest the CATCH capture site were chosen as input to determine the list of coexpressed genes via SEEK. Complete details can be found in Supplementary Table 2. The total number of genes in the SEEK data base is 17,857, and each final SEEK list used only the top 100/150 (based on coexpression value). In analyses where a random SEEK list was required (Fig. 5b–d), to determine the random distribution, the coexpression rankings of the SEEK gene list was permuted (at random) and its overlap was calculated with the CATCH-seq gene list. The lengths of the gene lists used to calculate the random was kept the same as the ones used to calculate the prediction by CATCH. Subsequently, the mean and s.d. of the random distribution was calculated and P value was determined using a t distribution. Only genes from the specific pull-down chromosome were used in this calculation. This was also considered optimal, as we were testing the viability of CATCH to predict gene coexpression, not the inverse. A flowchart detailing the steps of this analysis can be found in Fig. 5a, and a more detailed table of CATCH prediction of SEEK genes is available in Supplementary Table 3.

**Data analysis.** All R-codes are available upon request. All data analyses were performed using R version 3.2.2 within RStudio. Codes for all analyses are available upon request; however, the method will be covered briefly here. ERα ChIP-seq BED file data from MCF-7 cells (Fig. 2a–e) were obtained from ENCODE at the UCSC genome browser. Those data sets, from top-to-bottom were provided by the following: Barton, Brown, Carroll, Chinnaiyan, Hurtado (Carroll), Liu, Odom, Stunnenberg, Weisz and White laboratories. CATCH-seq data sets: Each CATCH-seq experiment was performed alongside an unfixed control experiment using an identical capture oligo. The raw data underwent FASTQ Groomer processing before being aligned to the hg19 build of the chromsome of interest using Bowtie 2. First, an unfixed control pulldown (CATCH experiment, minus any fixation method) using the same biotinylated oligonucleotide is normalized from the experimental pull-down data by aligning unfixed control reads (.bam) and 'subtracting' from the experimental reads, directly, to remove any background signal using the bamCompare function in Galaxy deepTools2. The data were then subsetted by individual chromosome (for example, chr3 for SIAH2, chr8 for MYC, and so on) in R, and then by signal threshold (the signal threshold is determined based on the number of CATCH interactions desired to discover). The identification of that threshold determined the signal strength at which CATCH peaks were defined as peaks; with the threshold determined, those CATCH peaks that satisfied the signal strength threshold in the BIGWIG of respective pulldown were next called. Next, gene promoters that had peaks within 2 k of their promoters were then annotated and considered as interactions with the pull-down locus. The top 500 (highest peaks) genes were then assessed to determine the three closest CATCH-identified genes to the pulldown. These three genes were used as the 'seed' for creating the SEEK list, which is described above. In the case of Supplementary Fig. 10, a continuous variable of peak numbers was discovered to determine the P value at which CATCH's ability to predict SEEK gene coexpression became nonsignificant. Enhancers: to determine the location of enhancers, ChIP-seq data from both H3K4me1 and H3K27ac were used. Peaks were called using the MACS (version 1.4.1; P value cutoff 1e − 05; MFOLD range 32, 128; fixed background lambda) function of Galaxy, and peak locations of at least 1 bp overlap between H3K4me1 and H3K27ac signal were identified. The combined distance of the two peaks was merged into a single peak, denoted an Enhancer, and made into a BED file for further use. ER binding: locations of ER binding were determined as above, instead using ERα ChIP-seq data and without combining any other data sets. CTCF binding: locations of CTCF binding were from CTCF-binding data from T47D cells (GEO accession: GSM803348). CATCH-seq gene identification: to ensure that only the most significant CATCH-seq peaks were used for analysis, gene promoters with peaks within 2 kb of their TSS were identified. If multiple peaks occurred near a TSS, only the most robust peak was considered. Then, the 500 genes with the strongest CATCH-seq signal peaks were identified for use in subsequent analyses. CATCH–enhancer adjacency/overlap: CATCH-seq signal strength (peak height) ranging from 100 to 500 was analysed. CATCH peaks were determined by filtering based on the strength of the BIGWIG files of the respective pulldown. Any CATCH-seq peak within 2 kb of an enhancer region (as defined above) was considered to be adjacent or overlapping, thus achieving our criteria for being considered an overlap in these analyses. TSS density plot: the plot(density(x)) function in R was used to plot CATCH-seq signal strength at locations within 2 kb up- and downstream of every TSS on a chromosome (chr3 for SIAH2, chr17 for EIF4A1 and chr8 for MYC). That signal density plot was used to determine the average location of signal 'peaks' near gene TSS. All R scripts are available in the online Supplementary Information as Supplementary Software.

**Data availability.** Previously published CTCF-binding data from T47D cells are available at the GEO under accession code GSM803348. The data that support the findings of this study are available on request from the corresponding author (G.L.G.; ggreene@uchicago.edu). The novel sequencing data from this study are available at the Gene Expression Omnibus (GEO), under accession code GSE85762. All original protocols related to CATCH and CATCH-seq library generation are also available upon request.

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

## Acknowledgements

We thank all members of the Greene Lab at the University of Chicago, especially Dave Hosfield, PhD, Sean W. Fanning, PhD, and Beth Russell, PhD for insightful discussions and manuscript edits. G.L.G. was funded by the Virginia and D.K. Ludwig Fund and NCI CA089489. R.J.B. was funded by the Susan G. Komen for the Cure postdoctoral fellowship and the University of Chicago Technological Innovation Fund.

## Author contributions

R.J.B. contributed funding, devised methods and experiments, carried out experiments, did bioinformatics and wrote the manuscript. H.S. created sequencing libraries and executed bioinformatics. G.L.G. provided funding, contributed critical scientific feedback and discussion, and edited the manuscript.

## Additional information

**Competing financial interests:** The authors declare no competing financial interests.

