## [Peer review file · Nature Communications]

Reviewers' Comments:

Reviewer #1 (Remarks to the Author)

This manuscript describes a novel method called CATCH which permits an unbiased method for DNA capture. CATCH is used to study ER enhancer elements, where known chromatin interaction regions are purified when a specific enhancer element is used as bait. Several loci are assessed using this approach and all produce chromatin interactions that reproduce known loops and in some cases, reveal new insight. The loops that form are shown to increase following stimulation of breast cancer cells with estrogen providing support for the hypothesis that this method captures active chromatin loops that form during transcription.

Overall this is a well-written manuscript and the method is likely to have wide applicability, but there are concerns about what it is that the authors are capturing and identifying.

- In multiple figures (i.e. Fig 2, 3 and 4), there are no read numbers on the Y-axis of the genomic tracks. As such, it is difficult to assess the efficiency of the CATCH method.
- It is unclear how many regions are pulled out with a specific CATCH experiment. The authors say they take the top 500 regions, but is this the top 500 regions on a single chromosome? If so this is a massive number that is at odds with every other 3C based discovery method. The authors need to make it clear how many total regions are captured.
- How dependent are these interactions on active transcription and does CATCH simply purify transcription factories? How many interactions (for a single region) are lost if transcription is blocked, potentially with alpha-amanitin?
- Given the large number of 'linked' chromatin regions, the reader assumes that there were interchromosomal interactions. Did the authors find chromatin loops between ER enhancer elements and different chromosomes? If so, can the authors speculate about why these were not seen with previous methods like Chia-PET?
- Figure 5 is not clear. Are the EIF4A captured regions the same as the MYC captured regions?
- Presumably these cells need estrogen to grow so is it correct that the 'vehicle' treatment could correlate with no growth? If so why are there only subtle differences between the estrogen and vehicle conditions in terms of the total number of captured genes?

Reviewer #2 (Remarks to the Author)

This review is focused on the authors' use of a human gene expression data search system, called SEEK, to validate whether the promoters captured by their method CATCH-seq are transcriptionally linked to each other for SIAH2-capture and 3 other loci capture experiments. To accomplish this evaluation, the authors mapped promoters (from CATCH-seq sequences) to the nearest genes and constructed an input list with 3 genes nearest to the probing loci's site. This input is entered as query into SEEK for identification of coexpressed genes. The authors showed that the coexpressed genes retrieved by SEEK overlap significantly with the gene promoters captured by CATCH-seq, thus implying that promoters are transcriptionally coordinated. The broader implication suggested by the authors is that enhancer may influence expression of many genes located far away on the chromosome, not just the gene closest to the enhancer.

This reviewer thinks that the use of coexpression to evaluate the transcriptional coordination of DNA interactions is reasonable. It is reasonable that the authors did not calculate coexpression based on a single gene expression dataset but rather utilized aggregated coexpression signatures from a large expression dataset collection. The coexpressed genes obtained this way should better serve to evaluate interactions across the broader biological contexts. So on a high level this analysis makes sense, but I have some specific concerns (see below).

Major points:

The CATCH-seq sequence data are not found in the supplementary files, making it impossible for me to review the data processing or any aspect of the data analyses performed (in the section Data Analysis in the Methods). Furthermore, SEEK results are not supplied in complete form (i.e. Supplementary Figure S8, see below) or not at all. For example, what is the complete list of top 100 coexpressed genes from SEEK that you use to calculate overlap for each capture experiment? This is not available in supplementary files, neither is the list of top 500 CATCH-seq genes.

The top 500 CATCH-seq genes are compared with SEEK, but it is not clear if the top 500 CATCH-seq genes span across all chromosomes or focus only on the chromosome where the probe-locus is located. This affects the genomic background used in calculating significance. It seems that the authors used a whole-genome background, since the SEEK genes were chosen from a background of 17,600 genes, the whole-genome. If so, this background seems to give an easy comparison, because interacting loci could be correctly selected just because they fall on the same chromosome as the probe-locus' chromosome. I would like to see a more stringent comparison between CATCH-seq and SEEK focusing only on the chromosome of the probe-locus, which would better highlight the ability to capture interacting loci within chromosome. This entails the use of all of 1400 genes on chromosome 8 as the random background for MYC-capture and 1600 genes on chr. 17 as background for EIF4A1-capture, etc, for both CATCH-seq and SEEK selected genes. Several studies have implicated CTCF/RAD21 as the proteins mediating promoter-enhancer looping (Ong et al, Nat. Rev. Genetics, 2014, 15, 234-246). CTCF creates boundaries for associated domains and facilitates interactions between regulatory sequences. I wonder if the CATCH-seq sequences overlap with CTCF or RAD21 binding sites across MCF-7 or T-47D cis-regulome and to what extent.

The reported overlap statistics are not backed by sufficient significance testing. As an example, what is the P value significance that 15 out of 100 genes overlap with CATCH-seq? All of the bars in the plots in Figure 5 should have P value significance indicated. None of them do currently. Robustness of the SEEK analysis should be improved. I would like to see authors repeat the overlap analysis (between SEEK and CATCH-seq) at other depths of SEEK coexpression ranking (i.e., not just top 100, but also include top 200 and 500 genes).

Other points:

Data analysis section in Methods is not detailed enough for others to replicate. In particular, CATCH-seq Data Sets (poorly written): "Furthermore, data analysis was done ... above a certain threshold", what threshold is that? What software did you use to process each CATCH-seq data set? How is sequencing data subtracted? How are peaks estimated from CATCH-seq?

Enhancers: what version of MACS is used? What parameters of MACS used? How wide are the peaks in general? Please attach MACS output for H3K4me1 and H3K27ac as supplementary files.

CATCH-seq gene identification: "Then the top 500 genes with the strongest CATCH-seq signal peaks..." - 500 genes on the same chr. or different chrs.? Out of how many genes are 500 selected?

CATCH/Enhancer overlap: "CATCH-seq signal strength ranging from 100 to 500..." what is unit for 100 and 500? "Any CATCH-seq peak within 2kb of an Enhancer region was considered to be overlapping". This is stretching definition of "overlapping". Is this even an overlap? I would name it as adjacent.

Creation of SEEK lists in Methods:

For all the input genes listed (ie MYC, TRIB1, FAM84B, SQLE, etc), I would also list their chromosome cytobands, such as MYC(8q24.21), TRIB1 (8q24.13), FAM84B (8q24.21) to indicate they are selected from nearby loci.

One of the input genes AADACP1 is a gene missing in SEEK. How did you perform the analysis in this case? Did you use alternative gene names for AADACP1? If so, this needs to be indicated.

In Suppl. Fig 8, I find that you only provide a partial list of genes retrieved by SEEK (even among the top 100), as the list does not match the SEEK website. This leads me to question if you had done post-processing with SEEK results. Please explain.

"To create each SEEK list, input genes were selected with the following rules" - Can you provide as supplementary file all input genes that follow these rules? You then proceed to use 3 of them as SEEK input for determining coexpressed genes. I would like to see complete list. In "for the chromosome 17 control...", indicate GRB7 gene being used as control.

In "(c) the gene must be unique to that experiment", it is not clearly defined what unique means. I understand the SIAH2 example given, but what does unique mean for the other two capture experiments (MYC, EIF4A1)?

Why is the SIAH2 SEEK list consisting of 150 genes, not 100 like the other capture experiments?

CATCH-seq predicts correlation with gene expression using the SEEK algorithm in Results section:

"SEEK determines correlation, over thousands of gene-expression data sets, between gene products." This is a misinterpretation of what SEEK does. SEEK calculates correlation between gene products by using a dataset-weighted correlation aggregation approach. The dataset weight is determined by the correlation of the input genes of interest. Because of dataset weighting, the correlation does not generalize over the entire collection of thousands of datasets (as would be implied by your writing), as it mostly depends on which datasets are up-weighted for a given input gene-set. The number of up-weighted datasets can range from many hundreds to a few thousands and only these datasets are being used for obtaining coexpressed genes. Please change this confusing sentence to emphasize dataset weighting aspect of SEEK. The SEEK analysis still holds; only the explanation should be changed.

Introduction: "It has become increasingly clear that the majority of ... but are typically located at great linear distance". What does linear distance mean? Longer-range perhaps?

Capture of Associated Targets on Chromatin (CATCH) identifies long-distance genomic interaction in Results: A figure or a table comparing the CATCH to the 3C,4C,5C, other tools and illustrating the differences would be nice to complement the text.

Figure 2: DNA strand orientation should be indicated for the genes LINC01213, SIAH2, ARHGEF26. As it stands, it looks like point A (called downstream enhancer of SIAH2 in the text) is actually located upstream of SIAH2 start site in this figure.

I find it nice that the SEEK input for the SIAH2-capture actually retrieves SIAH2 gene itself (in SEEK's coexpressed list), confirming the role of SIAH2 downstream enhancer in self-regulation. I wonder if the same can be true for the other capture experiments, MYC, EIF4A. Specifically, do the list of coexpressed genes from SEEK retrieve MYC, EIF4A themselves from using the respective SEEK inputs (which include nearby loci but not MYC EIF4A themselves)?

Reviewer #3 (Remarks to the Author)

In the manuscript, Bourgo and co-authors proposed a new method to detect chromatin interactions associated with a genomic locus of interest. Compared to the current 3C-based methods, the new method does not require enzyme digestion and ligation. Rather, sonication was used for genome fragmentation, and the biotinylated capture oligonucleotide was used for enriching the associated chromatin interactions. The results showed that the proposed methods could recover some chromatin interactions validated by previous methods and identify some new chromatin interactions. In general, the proposed method is an interesting design with potential advantages. However, considering the claims in the abstract, there are some concerns about the proposed method.

1. The experimental results are more like 4C experiments, which is widely-accepted method to capture "one-to-all" chromatin interactions. The authors should compare their proposed method with 4C, rather than a few examples from ChIA-PET, in order to demonstrate the efficiency and

reproducibility of the proposed method.

2. The authors mainly demonstrated their proposed method by examples around a few genes. What's the statistics for each experiment, such as the sequencing depth, the number of CATCH peaks, and the overlap rate with ER binding sites? With such statistics, it is easier to assess the method in general.

3. MCF7 is a cancer cell line with lots of structural variations and copy number variations. In order to interpret the experimental results properly, such variations should be considered. However, the variations in MCF7 are not mentioned in the manuscript.

4. In the METHODS part, "81{degree sign}C for 4 minutes (denaturation)" was mentioned in the protocol. By our understanding, such high temperature will affect the protein-DNA interactions - some interactions will be lost and chromatin interactions could NOT be detected. The authors need to estimate the effect of such high temperature.

And there are some minor concerns:

1. Page 4, paragraph 2, line 8: restriction enzyme digestion: it is not required in ChIA-PET for fragmentation.

2. Page 5, paragraph 2: "Furthermore, ER binding has not been detected within the SIAH2 promoter region, despite previous efforts to do so in MCF-7 cells 17." The ER binding data from Vega et al 2007 is out of date. Due to the technology limitation at that time, the coverage of ER binding data from Vega et al 2007 is quite low. Now, there are multiple ER ChIP-Seq data sets from MCF7. The authors should adopt the latest ER binding data sets.

3. page 8: "Importantly, without formaldehyde fixation, it was possible to pull down additional loci with ERE-B only if the sample was incompletely sonicated (Supplemental Figure S1B)." This is because ERE-B and ERE-A are close to each other in the linear genome. The interpretation here is inappropriate.

4. What's the Y-axis scale in Figure 2? Also for Figure 3 and 4.

5. Page 11: "Previous studies produced data suggesting that the promoter region of the human EIF4A1 gene in T47D cells is involved in multiple chromatin interactions with neighboring loci." What's the reference for the "previous studies"?

6. Page 17: "The current suite of assays available to probe DNA-DNA interaction are unreliable, time-consuming, difficult to use, and come with inherent flaws and difficulties that limit their practical effectiveness." The claim here, especially "unreliable", is inappropriate. If the authors claim the current methods are unreliable, how can they use ChIA-PET data to validate their method?

7. Page 20: "at the same within the same cellular space". "time" is probably missed after the first "same".

8. Page 24: "The resulting potential binding sites were then cross-referenced to previously identified ER-binding sites within MCF-7 cells 27." This sentence is contradicted with the sentence in Page 5, where ER binding sites from Vega et al 2007 were used.

9. Page 24: "(b) the peak near the TSS of the gene must be one of the top 500 (in height) such peaks on the chromosome". This criterion is not valid in MCF7 cells, since MCF7 cells have lots of structure variations and the copy numbers of different genomic fragments will affect the peak heights. In other words, the peak heights should be normalized with the local background.

10. "1% formaldehyde" is mentioned in the METHOD part, while "2% formaldehyde" is labeled in Figure 1. Which is the right concentration?

REVIEWER #1

Summary: Overall this is a well-written manuscript and the method is likely to have wide applicability, but there are concerns about what it is that the authors are capturing and identifying.

1. In multiple figures (i.e. Fig 2, 3 and 4), there are no read numbers on the Y-axis of the genomic tracks. As such, it is difficult to assess the efficiency of the CATCH method.

We agree that assay efficiency would be unclear without a scale; we have now updated our graph axes to indicate number of reads.

2. It is unclear how many regions are pulled out with a specific CATCH experiment. (a) The authors say they take the top 500 regions, but is this the top 500 regions on a single chromosome? (b) If so this is a massive number that is at odds with every other 3C based discovery method. (c) The authors need to make it clear how many total regions are captured.

(a) The sites identified in the current study are all on the single chromosome of interest, based on the pull-down. For instance, *SIAH2* is on chromosome 3, so all identified interactions are also on chromosome 3. The reason for our focus on intra- (as opposed to inter-) chromosomal interactions is given below, in the answer to point #4.

(b) This is an important but difficult point, because all of genomics hinges upon analysis thresholds. The CATCH analysis is similar to ChIP analysis, where changing the “threshold” by which you identify a protein binding site can result in the identification of 20,000 binding sites or 200 binding sites. Background and noise are subtracted, but algorithms based on thresholds ultimately determine what is called a “binding site” and what is ignored. Below, we did an analysis of half the number of CATCH genes on both of the *SIAH2* pull-down conditions.

Sample	No. CATCH Genes Identified	No. SEEK Genes Used	SEEK Genes Predicted at Random	SEEK Genes Predicted by CATCH	P-value
SIAH2-Veh	500	150	4.4	15	1.80E-08
SIAH2-E2	500	150	8	23	3.60E-13
SIAH2-Veh	250	150	4.4	14	4.10E-06
SIAH2-E2	250	150	7.8	15	0.008

As can be seen from this example, using only half the number of identified CATCH genes still results in a significant prediction of co-regulated genes via SEEK. Even after increasing the stringency of the SEEK p-value cutoff to 0.001, the same held true for EIF4A1 and MYC using 250 CATCH genes as well. Setting CATCH peak thresholds even higher (less identified genes) would potentially begin to push the limits of co-analysis with SEEK, as these combined analyses rely not only on the power of CATCH for the identification of DNA-DNA interactions, but also on the ability of SEEK to accurately determine co-regulated genes. One important consideration is that SEEK is using a weighted average based calculation over many different data sets, and that CATCH is the summation of DNA-DNA interaction across an entire population of cells. As such, larger numbers are required for statistical correlation between the two assays; attempting to identify (for instance) a single gene with CATCH and correlate that to findings from the SEEK database is essentially impossible. In regard to other 3C based discover methods, those techniques have focused largely on the number of interactions rather than number of genes associated with interactions at their promoters, so it is difficult to compare

CATCH with other techniques in this regard. Additionally, CATCH has vastly improved resolution (as noted in our new comparison of techniques in Table 1) compared to previous-generation techniques, as well as increased detection due to increased data capture (no loss of data via ligation), so it may be expected that CATCH discovers previously undetected interactions. Finally, other 3C-based techniques also employ thresholds within their data analyses, which can be changed to alter the number of “peaks” identified via those assays.

(c) In order to make it more clear how many and which genes were identified by CATCH (including all peaks identified for each gene, so the reader can assess relevance) we have now included complete lists for each experiment as Supplementary CSV files.

3. How dependent are these interactions on active transcription and does CATCH simply purify transcription factories? How many interactions (for a single region) are lost if transcription is blocked, potentially with alpha-amanitin?

This study has created a novel technique for the identification of chromosomal interactions and verified those DNA-DNA interactions through comparisons with established techniques. Additionally, our study has identified a novel pattern to those interactions by utilizing the SEEK database to interpret these data as correlated with transcription. However, we did not link them to *active, ongoing transcription*. Instead, we utilized a compendium of data (SEEK), which made possible a more global approach that allows for a high degree of statistical power. We were able to use Ingenuity Pathway Analysis (ex. Fig 4F) to demonstrate that the observed transcriptional patterns are biologically relevant. The reviewer is correct, however, that we cannot link our findings to active, ongoing transcription. We have now added additional discussion to clarify this point. We feel that transcriptional blockade experiments are beyond the scope of the current study.

4. Given the large number of 'linked' chromatin regions, the reader assumes that there were interchromosomal interactions. Did the authors find chromatin loops between ER enhancer elements and different chromosomes?

This is an excellent question. Unfortunately, we felt the depth (number of reads) of sequencing necessary to make claims about genome-wide interactions was beyond what was available to us in our core facility. Therefore, we analyzed only the single, most relevant chromosome for each CATCH pull-down (interactions mapping to the same chromosome as the pull-down). At present, we cannot comment about interchromosomal interactions.

5. Figure 5 is not clear. Are the EIF4A captured regions the same as the MYC captured regions?

The regions for each individual pull down are unique. The DNA-DNA interactions captured in these experiments were focused on single chromosomes (chromosome 17 for EIF4A1 and chromosome 8 for MYC), and thus are not overlapping. The former Figure 5 demonstrated the ability of CATCH (chromatin interaction) to predict gene co-expression (via SEEK) on a genome-wide scale, thus strengthening the hypothesis that genes that are co-expressed tend to physically interact. Due to the lack of clarity in this figure, as well as concerns by another reviewer, we have moved Figure 5 to the supplemental material, and carried out a completely new analysis to create a new Figure 5 that serves the same purpose. The new Figure 5 gives an overview of the new analytical process (Fig. 5A), and goes on to demonstrate that, for each CATCH pull-down (other than the *GRB7* control, Fig. 5B), the number of SEEK genes CATCH can predict on a specific chromosome is statistically significant compared to a random distribution. We feel that the single chromosome approach to this analysis is

more appropriate than the previous method and the statistical validation adds considerably to the conclusion that CATCH peaks correlate with SEEK co-expression results.

6. Presumably these cells need estrogen to grow so is it correct that the 'vehicle' treatment could correlate with no growth? If so why are there only subtle differences between the estrogen and vehicle conditions in terms of the total number of captured genes?

T47D cells divide in the absence of estrogen, although their doubling time is slower; thus, they are not completely growth inhibited in the absence of estrogen. Our current explanation for the [perhaps more modest than expected] difference in CATCH sites identified between vehicle and estradiol treatments, is a combination of static and plastic DNA-DNA interactions that seem to be utilized throughout the genome. Previous published research has identified this phenomenon, and we have added those references to the text as appropriate (Kulaeva *et al.*, *Mol Cell Biol*, 2012, and Kalhor *et al.*, *Nat Biotechnol*, 2012). Additionally, we are currently utilizing CATCH in a follow-up manuscript that deals with the recruitment of unliganded hormone receptors to the genome, which will shed more light on this topic.

REVIEWER #2

Summary: This reviewer thinks that the use of coexpression to evaluate the transcriptional coordination of DNA interactions is reasonable. It is reasonable that the authors did not calculate coexpression based on a single gene expression dataset but rather utilized aggregated coexpression signatures from a large expression dataset collection. The coexpressed genes obtained this way should better serve to evaluate interactions across the broader biological contexts. So on a high level this analysis makes sense, but I have some specific concerns (see below).

1. The CATCH-seq sequence data are not found in the supplementary files, making it impossible for me to review the data processing or any aspect of the data analyses performed (in the section Data Analysis in the Methods). Furthermore, SEEK results are not supplied in complete form (i.e. Supplementary Figure S8, see below) or not at all.

This was a glaring oversight on our part, but we have all of the relevant data files (as well as complete gene lists), which we will make available. The sequence data will be uploaded to GEO upon request for publication (as is customary). The complete gene lists (top co-expressed SEEK genes and top 500 CATCH-identified genes) are now incorporated into the supplementary data. We apologize for not providing these data originally. In addition, we have included more relevant processing information in the Data Analysis and Creation of SEEK lists sections in the materials in methods. We feel these changes have made it feasible to complete a reproduction of our data analysis.

2. I would like to see a more stringent comparison between CATCH-seq and SEEK focusing only on the chromosome of the probe-locus, which would better highlight the ability to capture interacting loci within chromosome.

This is a highly nuanced point, and an excellent question. As the reviewer noted, the SEEK list was created from a genome-wide list (approximately 17,600 genes), while our CATCH analysis was performed on a single chromosome of interest. When the top 100 SEEK genes were identified by the SEEK algorithm, *we did not filter out hits from other chromosomes*, which might appear counter-

intuitive for our analysis. As a result, the overlap analysis between SEEK and CATCH was actually biased against CATCH to some degree, because the SEEK list was allowed to identify genes from 22 other (irrelevant to our analysis) chromosomes that are impossible for CATCH to identify because we limited it to a single chromosome. Despite this, CATCH consistently identified far more genes from the SEEK list than were statistically expected at random. We do agree with the reviewer that chromosome-specific genes can be identified simply by virtue of being on the same chromosome as the pull-down. Therefore, we have now moved Figure 5 to the supplemental material and conducted a new analysis for an improved version of Figure 5 (see also Reviewer #1, point #5). In brief, the new Figure 5 uses SEEK genes exclusively from the same chromosome as the pull-down, and the random distribution calculated for those analyses are generated from the pool of genes available on SEEK, and only on the pull-down chromosome. This method removes any bias in the analysis and the p-values for each experiment are highly significant, as shown in red on the graphs.

3. Several studies have implicated CTCF/RAD21 as the proteins mediating promoter-enhancer looping (Ong et al, Nat. Rev. Genetics, 2014, 15, 234-246). CTCF creates boundaries for associated domains and facilitates interactions between regulatory sequences. I wonder if the CATCH-seq sequences overlap with CTCF or RAD21 binding sites across MCF-7 or T-47D cis-regulome and to what extent.

We have now done these analyses. We used CTCF-binding data from T47D cells (GEO accession: GSM803348) to reveal the percentage overlap between CTCF-binding and our various CATCH peaks (similar to those analyses seen in Figs. 3C and 4E). Supplemental Figure S10 now demonstrates that, for all pulldowns, CTCF has little or no significant binding overlap compared to random. We were somewhat surprised by the ‘negative’ result. CATCH is a technique that focuses on a single locus, and we specifically chose to look at loci impacted by estrogen receptor in this study. Our explanation is that while CTCF may be highly influential at particular sites of looping, it may not be associated as strongly with the particular interactions seen here.

4. The reported overlap statistics are not backed by sufficient significance testing. As an example, what is the P value significance that 15 out of 100 genes overlap with CATCH-seq? All of the bars in the plots in Figure 5 should have P value significance indicated. None of them do currently.

The updated version of Figure 5 now has p-values for each analysis, demonstrating statistical significance for all results except those of the *GRB7* control, as expected.

5. Robustness of the SEEK analysis should be improved. I would like to see authors repeat the overlap analysis (between SEEK and CATCH-seq) at other depths of SEEK coexpression ranking (i.e. not just top 100, but also include top 200 and 500 genes).

We agree that this analysis would strengthen the relationship between SEEK and CATCH, and have completed this analysis. The results are in Table 4 in the Supplemental materials, and demonstrate statistically significant CATCH prediction of SEEK results at all depths of SEEK analysis.

6. Data analysis section in Methods is not detailed enough for others to replicate. In particular, CATCH-seq Data Sets (poorly written): "Furthermore, data analysis was done ... above a certain threshold", what threshold is that? What software did you use to process each CATCH-seq data set? How is sequencing data subtracted? How are peaks estimated from CATCH-seq?

This is an excellent point. Our supplementary section now includes a fully detailed user protocol for

the experimental CATCH and self-made sequencing library preparation, and a more detailed methodology for the data processing flow using Galaxy and Galaxy DeepTools2 has been added to the materials and methods.

7. Enhancers: what version of MACS is used? What parameters of MACS used? How wide are the peaks in general? Please attach MACS output for H3K4me1 and H3K27ac as supplementary files.

MACS was version 1.4.1; the p-value cutoff for peak detection was $1e-05$, the MFOLD range was 32,128, and a fixed background lambda was used as local lambda for every peak region. This information is now included in the materials and methods. The average peak width was ~1200 for H3K4me1 and ~1500 for H3K27ac, and the BED files for H3K4me1 and H3K27ac will be uploaded to GEO and attached as supplemental files.

8. CATCH-seq gene identification: "Then the top 500 genes with the strongest CATCH-seq signal peaks..." - 500 genes on the same chr. or different chrs.? Out of how many genes are 500 selected?

This concern was also raised by Reviewer #1, point #2b. Please see above for more detail. Briefly, the 500 genes are on the same chromosome, and the total number of genes available is highly dependent upon thresholds (as it is with any similar assay like ChIP). When CATCH identified half as many genes (details in our response to Reviewer #1, point #2b), it was still able to significantly predict SEEK co-expressed genes.

9. CATCH/Enhancer overlap: "CATCH-seq signal strength ranging from 100 to 500..." what is unit for 100 and 500? "Any CATCH-seq peak within 2kb of an Enhancer region was considered to be overlapping". This is stretching definition of "overlapping". Is this even an overlap? I would name it as adjacent (page 27).

The units of measurement here are ‘number of sequencing reads after background subtraction’. This has now been noted on page 27, and in the figure legends, as suggested by the reviewer. Regarding adjacent versus overlapping, the reviewer’s suggested language is more accurate than what we stated, and we have changed the text to reflect that. We now say “adjacent or overlapping”, as our analysis actually covers both conditions.

10. For all the input genes listed (ie MYC, TRIB1, FAM84B, SQLE, etc), I would also list their chromosome cytobands, such as MYC(8q24.21), TRIB1 (8q24.13), FAM84B (8q24.21) to indicate they are selected from nearby loci.

This is an excellent suggestion, and we have now added a table (Table 2) to the Supplemental materials. This table details the “seed” genes, with corresponding cytobands, used to create the SEEK lists for each experiment, as well as the cytobands for each pull-down locus. The materials and methods section now references this table as well.

11. One of the input genes AADACP1 is a gene missing in SEEK. How did you perform the analysis in this case? Did you use alternative gene names for AADACP1? If so, this needs to be indicated.

This was a mistaken carry-over from a previous analysis and was overlooked during writing of the manuscript. The actual gene was supposed to be listed as *GYGI*, not *AADACP1*. We have changed it

to *GYGI* in the newly created Table 2, and any reference to *AADACPI* in the text has been removed (as mentioned above in comment #10). The original SEEK list used in the analysis is correct, however, and was actually created using *GYGI*, *IGSF10*, and *MBNLI*.

12. In Suppl. Fig 8, I find that you only provide a partial list of genes retrieved by SEEK (even among the top 100), as the list does not match the SEEK website. This leads me to question if you had done post-processing with SEEK results.

The data tables in Supplemental Figure S8 are genes from the SEEK list that also had significant CATCH-seq peaks within 2 kb of their transcription start sites. Because the SEEK list and CATCH-seq data do not overlap with 100% accuracy, the table shown in S8 is a subset of the total SEEK list. This information is now shown in bold lettering within Supplemental Figure S8 itself, in order to avoid any confusion. Additionally, as was addressed in point #1, above, complete gene lists for SEEK and CATCH are now available in the Supplemental Materials and Methods.

13. In "for the chromosome 17 control...", indicate GRB7 gene being used as control.

We have now labeled *GRB7* in each instance where chromosome 17 control was previously present in the figures and text.

14. "To create each SEEK list, input genes were selected with the following rules" - Can you provide as supplementary file all input genes that follow these rules? You then proceed to use 3 of them as SEEK input for determining coexpressed genes. I would like to see complete list. In "(c) the gene must be unique to that experiment", it is not clearly defined what unique means. I understand the SIAH2 example given, but what does unique mean for the other two capture experiments (MYC, EIF4A1)?

The confusion surrounding the “seed” list for SEEK led us to alter the text in the materials and methods for more clarity. As mentioned above, supplemental Table 2 includes all of the seed genes and their associated cytobands. Since only the 3 genes nearest to the pulldown by our calculations that fit this category were used, they would comprise the entirety of the list. Not accounting for distance-from-pulldown, many genes fit this list—the entirety of that CATCH-identified genes list is now available as a Supplementary CSV file.

15. Why is the SIAH2 SEEK list consisting of 150 genes, not 100 like the other capture experiments?

This explanation can be found, below, in answer to point #20.

16. "SEEK determines correlation, over thousands of gene-expression data sets, between gene products." This is a misinterpretation of what SEEK does. Please change this confusing sentence to emphasize dataset weighting aspect of SEEK. The SEEK analysis still holds; only the explanation should be changed.

The authors would like to thank the reviewer for such a clear and concise explanation; we have now updated our text to reflect his/her insight. The new text reads “SEEK determines gene expression correlation by weighting available gene expression datasets based on input genes of interest; using this weighted correlation aggregation method, it calculates relative gene co-expression amongst those datasets.”

17. "It has become increasingly clear that the majority of ... but are typically located at great linear distance". What does linear distance mean? Longer-range perhaps? (pg. 3, paragraph 2, line 9).

The reviewer is correct in his/her interpretation. The term is meant as longer-range (in base pairs). The issue was our misuse of the word “overwhelming” in: “Despite the overwhelming linear distance (in base pairs) between two interacting loci – in many cases, hundreds or thousands of kilobases...” . We have now, per the reviewer’s suggestion, replaced “overwhelming” with “long-range”. We think this will help to clarify the meaning of that statement.

18. A figure or a table comparing the CATCH to the 3C,4C,5C, other tools and illustrating the differences would be nice to complement the text.

We have now created this table, which can be found in the Supplemental Figures, Table 1.

19. Figure 2: DNA strand orientation should be indicated for the genes LINC01213, SIAH2, ARHGEF26. As it stands, it looks like point A (called downstream enhancer of SIAH2 in the text) is actually located upstream of SIAH2 start site in this figure.

This is an excellent suggestion and strand orientation is now shown in Figure 2.

20. I find it nice that the SEEK input for the SIAH2-capture actually retrieves SIAH2 gene itself (in SEEK's coexpressed list), confirming the role of SIAH2 downstream enhancer in self-regulation. I wonder if the same can be true for the other capture experiments, MYC, EIF4A. Specifically, do the list of coexpressed genes from SEEK retrieve MYC, EIF4A themselves from using the respective SEEK inputs (which include nearby loci but not MYC EIF4A themselves)?

An excellent point. This is the reason that 150 SEEK genes were used (for *SIAH2* as opposed to 100 for *MYC/EIF4A1*) to demonstrate the power of this analysis. *SIAH2* falls between numbers 100 and 150 on the SEEK list, thus we used 150 genes so that we could show that *SIAH2* validated itself. *Ultimately, whether we used 100, 200, 500 (or 17,000) SEEK genes, the statistics demonstrated that CATCH predicted more SEEK genes than statistically random. This is now illustrated in Supplemental Table 4.* Regarding *MYC* and *EIF4A1*, SEEK actually lists both of these genes within the top 15 genes (based on our 3 input genes for each). The *EIF4A1* pull down was within the promoter of the gene, so we could not identify its promoter via CATCH, because that was the pull down region. Regarding *MYC*, despite the fact that SEEK identified *MYC* as a gene co-regulated with genes identified by CATCH, our CATCH experiments did not identify the *MYC* promoter. This is not entirely unexpected, given that *MYC* is flanked by dozens of enhancer regions, and the interactions between *MYC*'s promoter and all of those enhancers is undoubtedly highly complex; it is possible that CATCH was either unable to capture this interaction, or the interaction was not present in the cell type or under the conditions we used in our experiments.

REVIEWER #3

1. The experimental results are more like 4C experiments, which is widely-accepted method to capture "one-to-all" chromatin interactions. The authors should compare their proposed method with 4C, rather than a few examples from ChIA-PET, in order to demonstrate the efficiency and reproducibility of the proposed method.

The reviewer is correct in saying that both 4C and CATCH are “one-to-all” capture techniques (this comparison can now be found in Supplemental Table 1). While we agree that 4C would be the optimal experiment if comparing the two methods directly, that was not our intention. Figure 2 arose from our original focus on the *SIAH2* enhancer locus, and the ChIA-PET experiments cited in this work not only looked at this locus broadly via sequencing, but also confirmed their findings directly via PCR. Similarly, we also confirmed our sequencing findings for this locus via PCR in Supplemental Figure S1. *The most critical aspect of Figure 2, however, was to confirm that CATCH was capable of detecting previously-identified DNA-DNA interactions, which was accomplished via both sequencing and PCR using the ChIA-PET data as our standard.* We have now included a table comparing various DNA interaction detection methods (Supplemental Table 2) which we feel makes critical comparison between many methods, as well as indicating that CATCH works in a distinct domain from 4C, thus not requiring a direct comparison. Another important factor is, to our knowledge, 4C-seq data for the *SIAH2* locus is not available, whereas the ChIA-PET data was published and validated.

2. The authors mainly demonstrated their proposed method by examples around a few genes. What's the statistics for each experiment, such as the sequencing depth, the number of CATCH peaks, and the overlap rate with ER binding sites? With such statistics, it is easier to assess the method in general.

The sequencing depth for each experiment was between 6 and 20 million reads; this is now noted in the materials and methods. The total number of CATCH peaks detected at or near promoters is discussed, above, in Reviewer #1 point #2b; please see that answer for full explanation. The overlap rate with ER binding sites can be found for each experiment in the new Supplemental Figure S9 (and for *SIAH2*-Vehicle in Supplemental Figure S7), which also looks at the potential overlap of CATCH peaks with CTCF binding. Statistics for CATCH and SEEK overlap are now included in the new Figure 5; full details about this can be found, above, in the answers to Reviewer #1 point #5, and Reviewer #2 point #4.

3. MCF7 is a cancer cell line with lots of structural variations and copy number variations. In order to interpret the experimental results properly, such variations should be considered. However, the variations in MCF7 are not mentioned in the manuscript.

The authors thank the reviewer for bringing this point to our attention. The cells used in Figure 2 for the validation of CATCH-seq (when compared to ChIA-PET) were MCF-7 at passage 3 after being purchased directly from the ATCC (HTB-22). According to the ATCC: cytogenetic analysis yielded a modal chromosome number of 82, with a range of 66 to 87. The stemline chromosome numbers ranged from hypertriploidy to hypotetraploidy, with the 2S component occurring at 1%. There were 29 to 34 marker chromosomes per S metaphase; 24 to 28 markers occurred in at least 30% of cells, and generally one large submetacentric (M1) and 3 large subtelocentric (M2, M3, and M4) markers were recognizable in over 80% of metaphases. No DM were detected. Chromosome 20 was nullisomic and X was disomic. This information is now supplied in the Materials and Methods section. The rest of the CATCH experiments were conducted in T-47D cells.

4. In the METHODS part, "81{degree sign}C for 4 minutes (denaturation)" was mentioned in the

protocol. By our understanding, such high temperature will affect the protein-DNA interactions - some interactions will be lost and chromatin interactions could NOT be detected. The authors need to estimate the effect of such high temperature.

This is a valid concern, and one that we tested during the creation of the CATCH protocol. In short, denaturation of 85°C or more significantly diminished the ability to detect DNA-DNA interactions via PCR. The current denaturation temperature in the protocol is 81°C, and via PCR, no noticeable loss of data was detected at this temperature. This is not to say that some data loss doesn't occur, but it was undetectable during testing. The critical balance is denaturing the DNA in order to allow open access for oligo annealing, without losing DNA-DNA interaction. Our testing revealed 81°C as an optimal temperature, as approaching ~75°C did not allow for sufficient oligo annealing in subsequent steps. Denaturation times over 5 minutes were not tested, so there are no data available describing how lengthier denaturation times impact this process. In sum, the denaturation step was crucial to the effectiveness of the protocol, and while this portion was optimized with PCR (not sequencing), that information is now available in the materials and methods section.

5. Page 4, paragraph 2, line 8: restriction enzyme digestion: it is not required in ChIA-PET for fragmentation.

This exception is now noted in the text and in the new Supplemental Table 1.

6. Page 5, paragraph 2: "Furthermore, ER binding has not been detected within the SIAH2 promoter region, despite previous efforts to do so in MCF-7 cells 17." The ER binding data from Vega et al 2007 is out of date. Due to the technology limitation at that time, the coverage of ER binding data from Vega et al 2007 is quite low.

This is not something we considered, but is absolutely correct. We have changed the text to reflect this point, and no longer cite Vega *et al.* to support this line of thinking.

7. page 8: "Importantly, without formaldehyde fixation, it was possible to pull down additional loci with ERE-B only if the sample was incompletely sonicated (Supplemental Figure S1B)." This is because ERE-B and ERE-A are close to each other in the linear genome. The interpretation here is inappropriate.

The reviewer is correct, and this explanation was originally intended, however it was poorly worded in the manuscript, leading to this confusion. We now added "due to the proximity of the two loci on the linear genome" to the explanation, in order to reflect the reviewer's comment.

8. What's the Y-axis scale in Figure 2? Also for Figure 3 and 4.

This is the same comment as reviewer #1 point #2; we have now given Y-axis labels to all genome browser graphs.

9. Page 11: "Previous studies produced data suggesting that the promoter region of the human EIF4A1 gene in T47D cells is involved in multiple chromatin interactions with neighboring loci." What's the reference for the "previous studies"?

This was an oversight on our part; we have now cited Li, G. *et al.* Extensive Promoter-Centered Chromatin Interactions Provide a Topological Basis for Transcription Regulation. *Cell* **148**, 84–98

(2012) in this location.

10. Page 17: "The current suite of assays available to probe DNA-DNA interaction are unreliable, time-consuming, difficult to use, and come with inherent flaws and difficulties that limit their practical effectiveness." The claim here, especially "unreliable", is inappropriate. If the authors claim the current methods are unreliable, how can they use ChIA-PET data to validate their method?

The reviewer is correct, our wording was problematic. The assays are not unreliable in terms of data quality; our original intent was to convey the concern (that many researchers express) for the number of times such experiments “fail”, requiring the experiment to be performed again until it produces useable data. Nonetheless, we have removed the word “unreliable”.

11. Page 20: "at the same within the same cellular space". "time" is probably missed after the first "same".

The reviewer was correct, the appropriate text was added.

12. Page 24: "The resulting potential binding sites were then cross-referenced to previously identified ER-binding sites within MCF-7 cells 27." This sentence is contradicted with the sentence in Page 5, where ER binding sites from Vega et al 2007 were used.

As discussed above in point 6, we have now removed the Vega *et al.* reference from that location, thus alleviating this potential contradiction.

13. Page 24: "(b) the peak near the TSS of the gene must be one of the top 500 (in height) such peaks on the chromosome". This criterion is not valid in MCF7 cells, since MCF7 cells have lots of structure variations and the copy numbers of different genomic fragments will affect the peak heights. In other words, the peak heights should be normalized with the local background.

The SEEK list creation was done only for experiments performed in T-47D cells. MCF-7 cells were only used in Figure 2 to validate CATCH’s ability to detect previously-identified interactions (from MCF-7 cells), but no correlation to SEEK was made in this figure. The text on page 26 has now been clarified to reflect this, so that this misunderstanding does not happen with the readers.

14. "1% formaldehyde" is mentioned in the METHOD part, while "2% formaldehyde" is labeled in Figure 1. Which is the right concentration?

This was an excellent catch by the reviewer. Both locations now correctly say 2% formaldehyde.

Reviewers' Comments:

Reviewer #1 (Remarks to the Author)

The authors have addressed my concerns. The new information and discussion clarifies any remaining issues I had.

Reviewer #2 (Remarks to the Author)

The manuscript has improved. The method section has been supplemented with more data analysis information. However a few issues remain:

1) In the response, the authors have clarified a major ambiguity in the manuscript, which is that the top 500 CATCH genes used for analysis are intra-chromosome, rather than selected across all chromosomes. However, with this information, one begins to wonder how well CATCH really works if you need to select so many genes to gain a significant result. 500 genes is over half of the chromosome! For the purpose of illustrating specificity of the technique, it appears that selecting the top 10% of probe chromosome's genes (i.e. top 100 genes) would be more appropriate. Although the authors said that "larger numbers are required for statistical correlation between the two assays", one could interpret that going to such large numbers as 500 genes is a sign of weak specificity of CATCH-seq. In order to be more convinced of the specificity of the technique, I would like to see analysis done on top 100, 150, 250 CATCH genes for all experiments: EIF4A1, MYC, SIAH2-Veh, SIAH2-E2 experiments (with p-value cutoff of 0.01 like authors have used in paper). So far, only SIAH2-Veh, SIAH2-E2 are provided for top 250 in authors response, but not others. The result for top 100 CATCH-seq genes should be the main results presented in the paper rather than the top 500.

2) Because the top X CATCH genes was so easily misinterpreted as inter-chromosomal in the original manuscript, I feel that the authors should make this clear in the main text. For example in the lines: "Then, the 500 genes with the strongest CATCH-seq signal peaks were identified for use in subsequent analyses." And "To test this hypothesis, the top 500 gene promoters for each CATCH-seq experiment were identified."

3) The authors have performed new analysis in response to my suggestion of picking genes from the pull-down chromosome to serve as background. I appreciate authors' effort. However, how this randomization is done or p-value calculated is not adequately described. For example, "the random SEEK list was determined by pulling an equal number of random gene names from the list of all possible SEEK genes available on the specific pull-down chromosome". This is confusing at first. What does equal number of random genes mean? Equal with what? Without clear explanation, the relevance of p-value generated by the authors in Figure 5 is much in doubt. Specifically, when determining a random SEEK list - one could shuffle only the ranks of the pull down chromosome genes keeping other gene ranks the same, then repeat the analysis. Or one could shuffle the ranks of all the genes regardless of which chromosome. Each leads to a very different p-value. It seems that authors should also try shuffling the CATCH list, as this eliminates the need to control for chromosome size & filtering if otherwise shuffling SEEK list.

4) The choice of MACS 1.4 (an older version of MACS, it is now updated to v2) seems unintuitive. Some people already discovered issues with MACS 1.4 <https://sites.google.com/site/anshulkundaje/projects/idr>. It is not clear if MACS v2 might give you an entirely different result, or if any other peak callers like HOMER, SPP, might give you different results than MAC v1.4 authors have used. I would like to see that the authors' analyses (such as comparison with SEEK) remain robust to peak calling methods.

5) With the now provided Supplementary Datasets (thanks for those), I was able to confirm that the MYC & SIAH2-e2 experiments had significant gene overlap against all depths of genes analyzed suggesting that there is good agreement between CATCH and coexpressed genes, and CATCH-selected genes can suggest transcriptional coordination. GRB7 was not significant as it should not be expected to. However, for EIF4A1 experiment particularly (either replicate), I was

worried that I could not get strong significance of overlap that the authors have reported (even for top 500 CATCH) - it was in fact not significant (see below for my calculation). Authors should fix this issue by perhaps clarifying the procedure of randomization & p-value as I am not able to generate P value as significant as reported in the paper Table 4.

EIF4A1: Genes on Chr17: 1019, Genes in SEEK in top 100 ($p=0.01$): 68, SEEK genes in CATCH top 500: 390, Overlap: 25. P-value (Fishers exact test): $p=0.65$.

Could the authors plot P-value vs. CATCH depth (100 to 500) and vs. SEEK depth (100 to 500) to show stability across different CATCH depths?

6) Authors have filtered SEEK's results based on $p=0.01$ cutoff, but in my experience, we generally use less stringent cutoff of $p=0.05$, as this should increase the number of genes for statistical comparisons. If the concern is lack of genes for reliable statistical comparison, maybe this is what authors should consider.

Reviewer #3 (Remarks to the Author)

The authors made great efforts to their work and now the manuscript is much improved. Still some questions are not clearly answered.

"What's the statistics for each experiment, such as the sequencing depth, the number of CATCH peaks, and the overlap rate with ER binding sites? With such statistics, it is easier to assess the method in general." The answer "The sequencing depth for each experiment was between 6 and 20 million reads" is ambiguous. The expected answers to the questions are the numbers to each experiment.

Furthermore, the original sequence data should be submitted to GEO for reviewer checking.

page 4: "Chromatin Conformation Capture (3C), as well as a number of derived techniques (e.g. 4C, 5C, Hi-C, ChIA-PET, T2C)", proper references are needed for different methods.

page 11: "Previous studies produced data suggesting that the promoter region of the human EIF4A1 gene in T47D cells is involved in multiple chromatin interactions with neighboring loci 19". The reference 19 (Li et al 2012) is inappropriate, since T47D cells were not used in Li et al 2012.

Some minor issues:

page 11, 4th-to-last line: spelling "threhold" should be "threshold"

Figure 5: Power of 10 in the p-values is not properly expressed.

REVIEWER #1

Concerns addressed; no additional comments.

REVIEWER #2

Summary: The manuscript has improved. The method section has been supplemented with more data analysis information. However a few issues remain.

1. In order to be more convinced of the specificity of the technique, I would like to see analysis done on top 100, 150, 250 CATCH genes for all experiments: EIF4A1, MYC, SIAH2-Veh, SIAH2-E2 experiments (with p-value cutoff of 0.01 like authors have used in paper).

It was demonstrated, statistically, that at multiple thresholds, CATCH-seq was predicting transcriptional co-expression, demonstrating the validity of CATCH-seq at those levels of specificity. The reviewer suggests attempting more specific analyses to test the ultimate resolution of CATCH, in terms of the prediction of transcriptional co-expression. *It is critical to remember that the stochastic nature of biology will limit the ability of a technique to predict a process like transcriptional co-expression with absolute certainty, especially when using thresholds that push the current technological limitations.* That said, we have now created a series of graphs that plot the relationship between the depths of the CATCH-seq gene lists and the significance of prediction of transcriptional co-expression. For various depths of SEEK list genes (100, 150, 200, and 300), the graphs demonstrate the p-value of each CATCH experiment as the number of CATCH genes captured (used) in the analysis decreases. A few highly interesting things to note come from this analysis. Firstly, the approximate resolution of CATCH's "SEEK gene predictive power" (denoted by the vertical red line; average point at which CATCH results become non-significant ($p=0.01$) of all CATCH experiments) increases with decreasing SEEK gene depth. This means that CATCH is more reliable at predicting the most significant SEEK genes, as might be expected with such complex biology. Secondly, while each CATCH experiment is slightly different, they all trend similarly, with the exception of the GRB7 experiment which is non-significant at all points, as expected from a negative control. At the original SEEK depth of 150 top SEEK genes, each CATCH experiment requires the following numbers of genes to significantly predict SEEK output:

EIF4A1-rep1: 145
EIF4A1-rep2: 153
MYC: 106
SIAH2 (E2): 183
SIAH2 (Vehicle): 107

Again, while these numbers vary, the effective "resolution" of CATCH falls between identifying about 100-200 genes. This effectively measures the resolution of the assay, not in base-pairs like ChIP, but in number of identified genes, making CATCH a technique that functions very well in predicting large cohorts of transcriptionally co-expressed genes, but less reliably for single locus-locus contacts.

These graphs have now been added as supplemental figures 11-14, which have the following Figure Legend(s):

Supplemental Figures S11-14: The graphs depict, using a constant SEEK list of the top 100, 150, 200, or 300 identified genes, the continuous p-value of each CATCH experiment as the number of CATCH genes identified (used) in the analysis decreases. The vertical dotted red lines indicate the average point (of all CATCH experiments on the graph) at which CATCH results become non-significant ($p=0.01$). As CATCH is used to identify more gene promoters, it's ability to predict transcriptionally co-expressed genes (as measured by SEEK) increases. The inverse \log_{10} of the p value was plotted on the y-axis against the number of CATCH genes used on the x-axis. The continuous p value was graphed for every CATCH experiment at varying steady-state SEEK depths of 100 (Figure S11), 150 (Figures S12), 200 (Figure S13) and 300 (Figure S14).

In addition, these findings are discussed in the discussion section, framed around the potential uses and limitations of the technique, as follows:

“The limitations of the CATCH assay are apparent, not from a base-pairs-of-resolution standpoint like ChIP, but seem to be more attached to the number of CATCH interactions needed to begin to statistically identify subsets of interacting genes. This point is illustrated in **Supplemental Figures S11-S14**, where, for various depths of SEEK list genes (100, 150, 200, and 300), the inverse \log_{10} of the CATCH p value (for SEEK gene prediction) was plotted on the y-axis against the number of CATCH genes identified in the analysis on the x-axis. Several interesting points arose from these analyses. Firstly, the approximate resolution of CATCH's transcriptional co-expression predictive power (as measured by SEEK list prediction), increases with decreasing SEEK gene depth. This effectively suggests that CATCH is more reliable at predicting the most significant SEEK genes, as might be expected with such complex biology. Secondly, while each CATCH experiment has slightly different effective resolution, they all trend similarly, with the exception of the GRB7 experiment which is non-significant at all points, as would be expected from a negative control. On average, the effective resolution of CATCH falls between identifying about 100-200 genes. This, to a degree, attempts to measure the resolution of the assay, not in base-pairs like ChIP, but in number of identified genes, making CATCH a technique that functions very well in predicting large cohorts of transcriptionally co-expressed genes, but less reliably for single locus-locus contacts. This, of course, does not dismiss any of the individual DNA-DNA interactions with CATCH; on the contrary, the ability of CATCH to identify physical interactions that statistically predict gene co-expression validates the technique. However, due to the complex nature of this biology, and the current technical limitations of detecting gene co-expression, it is statistically more likely to detect known DNA-DNA interactions when identifying a broader population of interactions with CATCH.”

2. Because the top X CATCH genes was so easily misinterpreted as inter-chromosomal in the original manuscript, I feel that the authors should make this clear in the main text. For example in the lines: "Then, the 500 genes with the strongest CATCH-seq signal peaks were identified for use in subsequent analyses." And "To test this hypothesis, the top 500 gene promoters for each CATCH-seq experiment were identified.

We have now added “on the chromosome of interest” to each of those statements, to clarify that we are identifying peaks only from the chromosome on which the experiment is being performed. However, to make the process completely clear within the text, we have now added the following information to the Materials and Methods Section:

“Firstly, an unfixed control pulldown (CATCH experiment, minus any fixation method) using the same biotinylated oligonucleotide is subtracted from the experimental pulldown data. The aligned unfixed control reads (.bam) were ‘subtracted’ from the experimental reads, directly, to remove any background signal using the bamCompare function in Galaxy deepTools2. The data was then subsetted by individual chromosome (e.g. chr3 for *SIAH2*, chr8 for *MYC*, etc.), in R, and then subsetted by signal threshold (the threshold is auto-set based on the number of interactions desired to discover). The identification of that threshold determines the signal strength at which peaks are determined, and gene promoters that have peaks within 2k of their promoters are then annotated and considered as interactions with the pulldown locus. The top 500 (highest peaks) genes were then assessed to determine the three closest CATCH-identified genes to the pull-down. These three genes were used as the “seed” for creating the SEEK list, which is described above. In the case of Supplemental Figures 11-14, a continuous variable of peak numbers was discovered to determine the p value at which CATCH's ability to predict SEEK gene co-expression became non-significant.”

3. The authors have performed new analysis in response to my suggestion of picking genes from the pull-down chromosome to serve as background. I appreciate authors' effort. However, how this randomization is done or p-value calculated is not adequately described.

This randomization process is more clearly defined in the Materials and Methods section. While the reviewer

has listed a number of ideas for calculating randomization here, we believe that the issue stems from our poor explanation of how random was calculated. The following text was added to the materials and methods:

“To determine the random distribution, the co-expression rankings of the SEEK gene list was permuted and its overlap was calculated with the CATCH-seq gene list. The lengths of the gene lists used to calculate the random was kept the same as the ones used to calculate the prediction by CATCH. Subsequently, mean and standard deviation of the random distribution was calculated and p value was determined using a t distribution”.

It is an optimal way to calculate random in this instance (randomizing the SEEK list); randomizing the CATCH peaks doesn't allow us to test the validity of CATCH to predict co-expression. We must randomize the SEEK list to validate whether CATCH can predict gene co-expression from those “real” or “fake” SEEK lists.

4. The choice of MACS 1.4 (an older version of MACS, it is now updated to v2) seems unintuitive. Some people already discovered issues with MACS 1.4. It is not clear if MACS v2 might give you an entirely different result, or if any other peak callers like HOMER, SPP, might give you different results than MAC v1.4 authors have used. I would like to see that the authors' analyses (such as comparison with SEEK) remain robust to peak calling methods.

MACS was not used to detect the CATCH peaks, only the ChIP-seq peaks. Identification of CATCH-seq peaks was done in R; this information is in the Materials and Methods:

“CATCH-seq signal strength (peak height) ranging from 100 to 500 was analyzed. CATCH peaks were determined by filtering based on the strength of the signal in BIGWIG files of the respective pull-down. Any CATCH-seq peak within 2kb of an Enhancer region (as defined above) was considered to be adjacent or overlapping, thus achieving our criteria for being considered an overlap in these analyses. TSS Density Plot: the plot(density(x)) function in R was used to plot CATCH-seq signal strength at locations within 2kb up- and downstream of every TSS on a chromosome (chr3 for SIAH2, chr17 for EIF4A1, and chr8 for MYC). That signal density plot was used to determine the average location of signal “peaks” near gene TSS.”

5. With the now provided Supplementary Datasets (thanks for those), I was able to confirm that the MYC & SIAH2-e2 experiments had significant gene overlap against all depths of genes analyzed suggesting that there is good agreement between CATCH and coexpressed genes, and CATCH-selected genes can suggest transcriptional coordination. GRB7 was not significant as it should not be expected to. However, for EIF4A1 experiment particularly (either replicate), I was worried that I could not get strong significance of overlap that the authors have reported (even for top 500 CATCH) - it was in fact not significant (see below for my calculation). Authors should fix this issue by perhaps clarifying the procedure of randomization & p-value as I am not able to generate P value as significant as reported in the paper Table 4.

We are pleased that the reviewer was able to repeat some of our analysis with success, thus demonstrating that CATCH-seq was capable of identifying biologically co-expressed genes. We believe that this particular concern is similar in nature to concern #1, and the analyses/graphs created in response to that concern should satisfy the reviewer's request for various combinations of CATCH-seq and SEEK thresholds. In addition, we have now clarified our process of determining random, and have demonstrated that the method we use is optimal in this situation, as mentioned in our response to concern #3.

We are unsure why the Reviewer has not been able to replicate our findings with the *EIF4A1* CATCH experiment, but the new supplemental figures should put the reviewer's concerns at ease, as we have calculated the p-value for every CATCH experiment at every threshold on a continuous graph.

6. Authors have filtered SEEK's results based on p=0.01 cutoff, but in my experience, we generally use less stringent cutoff of p=0.05, as this should increase the number of genes for statistical comparisons. If the concern is lack of genes for reliable statistical comparison, maybe this is what authors should consider.

This request seems counter-intuitive; it seems unnecessary to make our analysis less specific by reducing p-value cutoff stringency to $p \leq 0.05$, when our current analysis uses $p \leq 0.01$. While this may increase the number of genes for statistical comparisons, this would be a less precise method than decreasing the threshold of CATCH data to include a larger number of identified genes, and would not help to determine technique specificity (the primary concern of the reviewer).

Instead, we have continued our analysis at $p=0.01$, and our new Supplemental Figures 11-14 demonstrate the resolution of CATCH for prediction of transcriptional co-expression at all meaningful levels.

REVIEWER #3

Summary: The authors made great efforts to their work and now the manuscript is much improved. Still some questions are not clearly answered.

1. "What's the statistics for each experiment, such as the sequencing depth, the number of CATCH peaks, and the overlap rate with ER binding sites? With such statistics, it is easier to assess the method in general." The answer "The sequencing depth for each experiment was between 6 and 20 million reads" is ambiguous. The expected answers to the questions are the numbers to each experiment.

Regrettably, we gave the reviewer less information than they felt necessary, and our original range was mistaken. We have now added specific read numbers for each experiment to the Materials and Methods section, as follows:

“A complete user protocol is available in the Supplemental Materials; sequencing depth for each library varied between ~15 to ~24 million reads: GRB7 replicates 1 and 2 had 16.0 and 22.9 million reads, respectively, MYC had 16.1 million reads, EIF4A1 replicates 1 and 2 had 18.6 and 24.2 million reads, respectively, SIAH2 vehicle-treated had 16.2 million reads, and SIAH2 estradiol-treated had 15.1 million reads.”

2. Original sequence data should be submitted to GEO for reviewer checking.

This will be completed upon acceptance.

3. Page 4: "Chromatin Conformation Capture (3C), as well as a number of derived techniques (e.g. 4C, 5C, Hi-C, ChIA-PET, T2C)", proper references are needed for different methods.

We have now altered the text; each of these techniques is discussed in the discussion section with its proper original reference.

4. Page 11: "Previous studies produced data suggesting that the promoter region of the human EIF4A1 gene in T47D cells is involved in multiple chromatin interactions with neighboring loci". The reference 19 (Li et al 2012) is inappropriate, since T47D cells were not used in Li et al 2012.

We have now reworded this to say “Previous studies produced data suggesting that the promoter region of the human EIF4A1 gene is involved in multiple chromatin interactions with neighboring loci.” This removes the conflict with the reference.

5. Page 11, 4th-to-last line: spelling "threhold" should be "threshold"

We have changed to the correct spelling.

6. Figure 5: Power of 10 in the p-values is not properly expressed.

The exponents are now expressed as superscript.

Reviewers' Comments:

Reviewer #2 (Remarks to the Author)

The new analyses are satisfactory. In general, I agree with the authors that CATCH is appropriate in identifying large cohorts of coexpressed genes.

However, the explanations added to the discussion contains few points of confusion, which I suggest revising. Please see below:

"This, to a degree, attempts to measure the resolution of the assay, not in base-pairs like ChIP, but in number of identified genes, making CATCH a technique that functions very well in predicting large cohorts of transcriptionally co-expressed genes, but less reliably for single locus-locus contacts."

I think it is confusing and contradictory to say that it is less reliable at single locus contact, while also saying that it does not dismiss any individual DNA interactions (predicted) with CATCH.

"This, of course, does not dismiss any of the individual DNA-DNA interactions with CATCH; on the contrary, the ability of CATCH to identify physical interactions that statistically predict gene co-expression validates the technique. However, due to the complex nature of this biology, and the current technical limitations of detecting gene co-expression, it is statistically more likely to detect known DNA-DNA interactions when identifying a broader population of interactions with CATCH"

When identifying a broader populations of interactions, one can also say that it is increasingly likely to identify more false interactions, in addition to identifying known DNA-DNA interactions. The reviewer thinks this sentence is a little confusing, and may benefit from the below additions the reviewer proposes.

"On average, the effective resolution of CATCH falls between identifying about 100-200 genes." Could authors clarify what resolution of CATCH means? Perhaps authors can consider these additions I propose: Going below the number of genes in effective resolution may not allow CATCH to identify statistically significant number of coexpressed genes, as it may be easily affected by noises, variations, complex nature of this biology. Interpretations of single gene locus-probe interaction should proceed with caution. Nonetheless, by summarizing interactions to hundreds of identified genes, this analysis illustrates the collective effect of interactions in mediating gene coexpressions.

The Method section has improved, but the reviewer still think it can improve in terms of providing more details to the users wishing to repeat the analysis (see details below)

Other revisions:

Line 91: "Estrogen Receptor", should be "estrogen receptor (ER)", or simply ER.

93-94: SEEK algorithm - we recommend calling it SEEK search system

113-114: I spent a lot of time looking at this Supp Fig S1B. The legend is confusing: "However, with complete sonication and fixation, the ERE downstream of SIAH2 is also captured with the ERE-B". There are two panels that are complete sonication and fixation. Which is it? I assume it is one in ERE-B capture panel. If so, please refer to it in the legend. Also in the same sentence, can you specify examples of the ERE's downstream of SIAH2 that I should expect to see in the gel? Is one example of ERE downstream the gene DNSTRM?

Figure 2: there are largely two sections: (top), (A-E). This is unconventional way to name the sections. I recommend either use (top), (bottom), or rename top as (A), and the rest needs to be renamed as well. This figure also causes much confusion. Without any clarification, it took me a while to figure out that A-E (the section titles) actually refers to the 5 locus points in the top panel of the same figure. If so, please put an explanatory sentence in legend or text.

151,154,157: I think you are referring to Figure 2A, 2B, 2C instead of 3A-C.

164: "An overview comparison of CATCH..." - awkward to mention this here, does not flow well.

213: What are the P-values of the processes identified by Ingenuity Pathway Analysis? Include p-values.

Fig 3E, 4F: gene names not legible.

408: sentence not clear. "Inverse log10" - I think you mean negative log10? Same goes for the Figures (Suppl. Fig S11-14). What does CATCH p value mean?

Fig 5: Combine steps E and F into something like: subset into top coexpressed genes from SEEK and on pull-down chromosome. Rename step G as: compute overlaps and statistical significance with CATCH identified genes (from B)

Methods revision:

585: we suggest depositing code in Github or Bitbucket

595: "subtracted" - poor word choice, suggest "corrected". How is subtraction done? Place reference. The next sentence seems to answer this question, and therefore redundant. Take either sentence, remove the other.

598: two mentions of word "subsetting", remove one to combine into a tighter sentence: "the data was then subsetted by individual chromosome, and by signal threshold."

599: how is threshold auto-set? what are number of interactions you have covered? you could say something like: "threshold is set based on achieving X interactions (... experiment), Y interactions (...experiment)..." . How did you determine # interactions desired? Based on a fixed percentage?

600: "the identification of that threshold..." sentence is redundant, as it is obvious signal threshold is used to filter signal strength. suggest removal.

601: "CATCH peaks were called simply by" - revise this as: "with the threshold determined, those CATCH peaks that satisfied the signal strength threshold in the BIGWIG of respective pull-down were next called."

Reviewer #3 (Remarks to the Author)

My concerns have been addressed.

REVIEWER #2

1. The reviewer requests additions to the discussion to clarify the main limitation of CATCH.

We have now added the following text to the discussion, following very closely the original wording proposed by the reviewer, in order to clarify the main limitation of CATCH:

Attempting to identify fewer genes than the effective resolution of CATCH may not result in the identification of statistically significant co-expressed genes, as it may be impacted by noise and the variations inherent to the complex nature of this biology. Interpretations of single gene locus-probe interaction should be used with caution. Nonetheless, by summarizing interactions to hundreds of identified genes, this analysis illustrates the collective effect of interactions in mediating gene co-expression.

2. Change Estrogen Receptor to “estrogen receptor (ER)” – line 91.

Change made.

3. Change SEEK algorithm to “SEEK search system” – line 94.

Change made.

4. Change Supplemental Figure S1B legend to be more clear.

Change made; the legend now refers to EREA and DNSTRM to illustrate which fragments are being captured in each panel.

5. Change Figure 2 legend to be more self-explanatory.

Change made; the legend now refers to **Top Diagram** and **Sub-Panels**, prior to actually delineating each of the panels (A, B, C, etc). We feel this change is sufficient to make the figure legend clear.

6. Instance of citing Figures 3A, 3B, 3C, and 3D should have actually been referring to Figure 2A, B, C and D.

The reviewer was correct, this was much appreciated. The change was made.

7. Reviewer suggests that citing Supplemental Table 1 was awkward where it was located in the text.

We agreed, and moved this citation/reference to the Methods section.

8. Figure 3E and F4 gene names not legible.

We have included a higher resolution version of the main figures so that whichever version is best for the journal should be published.

9. Change inverse log₁₀ to negative log₁₀, and explain p-value for CATCH

Both were done; the CATCH p-value explanation is now in the figure legend for Supplemental Figures 11-14, where appropriate.

10. Alter methodology from subtraction, to normalization – line 595.

Done.

11. Remove one instance of the word subsetted – line 598.

Done.

12. Change wording around the “auto-setting” of threshold – line 599.

We have updated the sentence, particularly to remove the term “auto-set”, but we feel the sentence is otherwise clear. Especially with the publication of our R codes, which we will upload to protocol exchange with our CATCH protocol.

13. Change wording of line 601.

Wording changed to be less redundant, and to satisfy the reviewer's requirement for additional clarity:

The identification of that threshold determined the signal strength at which CATCH peaks were defined as peaks; with the threshold determined, those CATCH peaks that satisfied the signal strength threshold in the BIGWIG of respective pull-down were next called.